# European and multi-ancestry genome-wide association meta-analysis of atopic dermatitis highlights importance of systemic immune regulation

Atopic dermatitis (AD) is a common inflammatory skin condition and prior genome-wide association studies (GWAS) have identified 71 associated loci. In the current study we conducted the largest AD GWAS to date (discovery N = 1,086,394, replication N = 3,604,027), combining previously reported cohorts with additional available data. We identified 81 loci (29 novel) in the European-only analysis (which all replicated in a separate European analysis) and 10 additional loci in the multi-ancestry analysis (3 novel). Eight variants from the multi-ancestry analysis replicated in at least one of the populations tested (European, Latino or African), while two may be specific to individuals of Japanese ancestry. AD loci showed enrichment for DNAse I hypersensitivity and eQTL associations in blood. At each locus we prioritised candidate genes by integrating multi-omic data. The implicated genes are predominantly in immune pathways of relevance to atopic inflammation and some offer drug repurposing opportunities.

Atopic dermatitis (AD, or eczema) is a common allergic disease, characterised by (often relapsing) skin inflammation affecting up to 20% of children and 10% of adults[1]. Several genome-wide association studies (GWAS) have been performed in recent years, identifying genetic risk loci for AD.

Our most recent GWAS meta-analysis within the EAGLE (EArly Genetics and Lifecourse Epidemiology) consortium, published in 2015 uncovered 31 AD risk loci[2]. Since then, additional GWAS have been published which have confirmed known risk loci[3,4] and discovered novel loci[5]. Five novel loci were identified in a European meta-analysis[6], and variants in 3 genes were implicated in a rare variant study in addition to 5 novel loci[7]. Four novel loci were reported in a Japanese population (and another 4 identified in a trans-ethnic meta-analysis in the same study)[8], giving a total of 71 previously reported AD loci[2–14] (defined as 1 Mb regions) of which 57 have been reported in European ancestry individuals, 18 have been reported in individuals of non-European ancestry and 29 in individuals across multiple ancestry groups (Supplementary Data 1).

The availability of several new large population-based studies has provided an opportunity to perform an updated GWAS of AD, aiming to incorporate data from all cohorts that have contributed to previously published AD GWAS, as well as data from additional cohorts, to present the most comprehensive GWAS of AD to date, including comparison of effects between European, East Asian, Latino and African ancestral groups. In this work we identify novel loci and use multi-omic data to further characterise these associations, prioritising candidate causal genes at individual loci and investigating the genetic architecture of AD in relation to tissues of importance and shared genetic risk with other traits.

## Results

### European GWAS

The discovery European meta-analysis (N = 864,982; 60,653 AD cases and 804,329 controls from 40 cohorts, summarised in Supplementary Data 2) identified 81 genome-wide significant independent associated loci (Fig. 1a and Supplementary Fig. 1). 52 were at

e-mail: l.paternoster@bristol.ac.uk

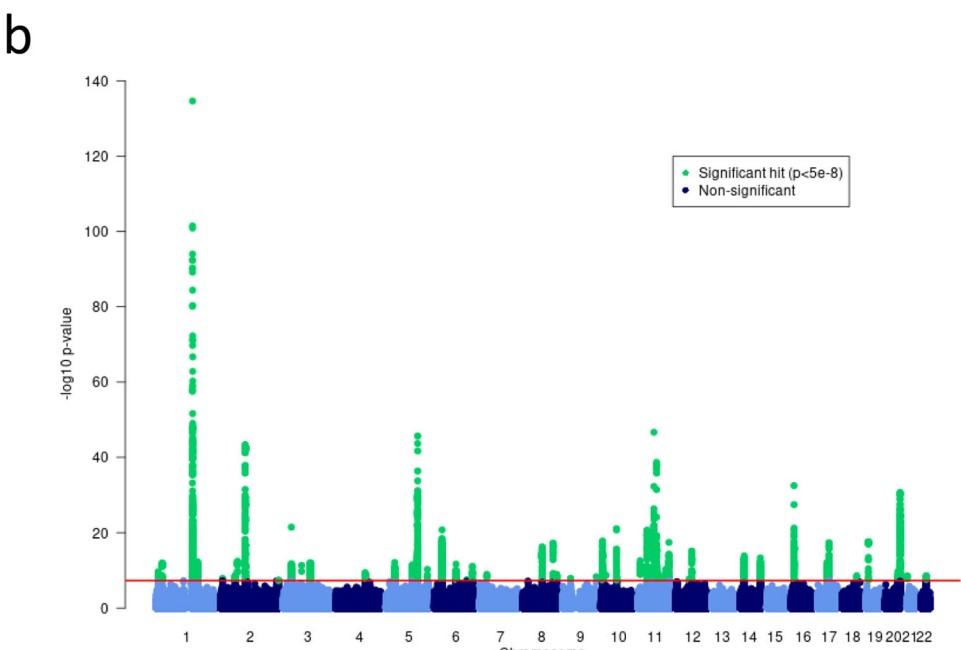

**Fig. 1 | Manhattan plots of atopic dermatitis GWAS.** (**a**) the European-only fixed effects meta-analysis (*n* = 864,982 individuals) and (**b**) the multi-ancestry MR-MEGA meta-analysis (*n* = 1,086,394 individuals). −log$_{10}$(*P*-values) are displayed for all variants in the meta-analysis. Variants that meet the genome-wide significance threshold (5 × 10$^{-8}$, red line) are shown in green.

previously reported loci (Table 1) and 29 (Table 2) were novel (according to criteria detailed in the methods). All 81 were associated in the European 23andMe replication analysis (Bonferroni corrected *P* < 0.05/81 = 6 × 10$^{-4}$, *N* = 2,904,664, Table 1). There was little evidence of genomic inflation in the individual studies (lambda <1.05) and overall (1.06). Conditional analysis determined 44 additional secondary independent associations (*P* < 1 × 10$^{-5}$) across 21 loci (Supplementary Data 3).

The SNP-based heritability ($h^2_{SNP}$) for AD was estimated to be 5.6% in the European discovery meta-analysis (LDSC intercept=1.042 (SE = 0.011)). This is low in comparison to heritability estimates for twin studies (~80%)[15,16], but comparable with previous $h^2_{SNP}$ estimates for AD in Europeans (5.4%)[6].

## Multi-ancestry GWAS
In a multi-ancestry analysis including individuals of European, Japanese, Latino and African ancestry (Supplementary Data 2, *N* = 1,086,394; 65,107 AD cases and 1,021,287 controls), a total of 89 loci were identified as associated with AD (Fig. 1b and Supplementary Fig. 1). 75 of these were not independent of lead variants identified in the European-only analysis ($r^2$ > 0.01 in the relevant ancestry) and a further 9 showed some evidence for association (Bonferroni corrected *P* < 0.05/89 = 5.6 × 10$^{-4}$) in the European analysis, but 5 were not associated (*P* > 0.1) in Europeans (Table 3, Supplementary Data 4).

Of the 14 loci that reached genome-wide significance in the multi-ancestry discovery analysis only (Table 3), 8 replicated in at least one of

**Table 1 | Genome-wide significant loci in European-only analysis that have been previously reported**

| Variant | Chr:position | Alleles (EAF) | European discovery | | | Multi-ancestry discovery | | 23andMe European replication (N = 2,904,664) | | Gene | Pathway/Function |
|---|---|---|---|---|---|---|---|---|---|---|---|
| | | | OR (CI) | P | N (studies) | P | N (studies) | OR (CI) | P | | |
| rs7542147 | 1:25294618 | C/T (0.49) | 1.04 (1.03-1.06) | 8.52E-11 | 860840 (38) | 2.4E-09 | 870216 (42) | 1.05 (1.04-1.05) | 4.6E-56 | RUNX3 | Versatile transcription factor, incl. T cell differentiation |
| rs12123821 | 1:152179152 | T/C (0.05) | 1.40 (1.35-1.45) | 4.05E-90 | 850727 (29) | 2.3E-98 | 857207 (31) | 1.27 (1.25-1.29) | 1.4E-228 | FLG | Skin barrier protein |
| rs6816766[a] | 1:152319572 | C/T (0.03) | 1.66 (1.58-1.74) | 6.44E-89 | 627936 (20) | 1.1E-102 | 634416 (22) | 1.41 (1.39-1.43) | 1.4E-228 | FLG | Skin barrier protein |
| rs72702900 | 1:152771963 | A/T (0.04) | 1.28 (1.24-1.33) | 2.98E-46 | 851612 (29) | 3.0E-49 | 853748 (30) | 1.23 (1.22-1.25) | 4.2E-163 | FLG | Skin barrier protein |
| rs61815704 | 1:152893891 | G/C (0.02) | 1.78 (1.67-1.89) | 3.21E-71 | 530473 (19) | 9.2E-72 | 536953 (21) | 1.36 (1.34-1.39) | 5.5E-212 | S100A9[b] | TLR4 signalling |
| rs12133641 | 1:154428283 | G/A (0.39) | 1.07 (1.05-1.08) | 1.72E-21 | 857974 (37) | 1.8E-22 | 1079390 (42) | 1.04 (1.04-1.05) | 3.0E-45 | IL6R | Cytokine signalling in immune system |
| rs859723 | 1:172744543 | A/G (0.36) | 0.94 (0.93-0.96) | 3.74E-14 | 522713 (37) | 2.4E-14 | 744125 (42) | 0.96 (0.96-0.97) | 2.2E-39 | TNFSF4[b] | Cytokine signalling in immune system |
| rs11811788 | 1:173150727 | G/C (0.24) | 1.07 (1.05-1.08) | 1.85E-17 | 859747 (38) | 3.1E-16 | 1081160 (43) | 1.04 (1.04-1.05) | 1.6E-39 | TNFSF4 | Cytokine signalling in immune system |
| rs891058 | 2:8442547 | A/G (0.29) | 0.96 (0.94-0.97) | 1.76E-10 | 862482 (38) | 2.2E-11 | 1083890 (43) | 0.97 (0.97-0.98) | 3.0E-18 | ID2 | Transcriptional regulator of many cellular processes |
| rs112111458 | 2:71100105 | G/A (0.12) | 0.94 (0.92-0.96) | 5.50E-09 | 858567 (37) | 1.4E-11 | 1079980 (42) | 0.96 (0.95-0.97) | 1.3E-21 | CD207 | Dendritic cell function |
| rs2272128 | 2:103039929 | A/G (0.77) | 0.91 (0.90-0.92) | 8.14E-35 | 862259 (39) | 3.8E-48 | 1083670 (44) | 0.93 (0.93-0.94) | 2.2E-100 | IL18RAP | Cytokine signalling in immune system |
| rs4131280 | 3:18414570 | A/G (0.57) | 0.96 (0.95-0.98) | 1.2E-08 | 864982 (40) | 5.8E-08 | 1086390 (45) | 0.97 (0.97-0.98) | 2.2E-19 | SATB1 | Regulates chromatin structure and gene expression |
| rs13097010 | 3:18673161 | G/A (0.34) | 1.05 (1.03-1.06) | 9.0E-11 | 864982 (40) | 1.5E-08 | 1086390 (45) | 1.02 (1.01-1.02) | 1.4E-07 | SATB1 | Regulates chromatin structure and gene expression |
| rs35570272 | 3:33047662 | T/G (0.40) | 1.04 (1.03-1.05) | 5.7E-09 | 864982 (40) | 2.3E-20 | 1086390 (45) | 1.03 (1.03-1.04) | 1.6E-26[a] | GLB1 | Sphingolipid metabolism |
| rs6808249 | 3:112648985 | T/C (0.54) | 0.96 (0.95-0.97) | 9.05E-11 | 859747 (38) | 3.8E-12 | 1081160 (43) | 0.97 (0.96-0.97) | 4.7E-29 | CD200R1 | Adaptive immune system |
| rs45599938 | 4:123386720 | A/G (0.35) | 1.05 (1.03-1.06) | 4.61E-12 | 859747 (38) | 3.7E-10 | 1081160 (43) | 1.05 (1.05-1.06) | 1.3E-62 | KIAA1109 | Endosomal transport |
| rs10214273 | 5:35883986 | G/T (0.27) | 0.94 (0.93-0.96) | 5.97E-16 | 863209 (39) | 1.8E-14 | 1084620 (44) | 0.93 (0.93-0.94) | 2.9E-99 | IL7R | Cytokine signalling in immune system |
| rs17132590 | 5:110331899 | C/T (0.10) | 1.07 (1.05-1.10) | 1.16E-08 | 525225 (38) | 1.7E-08 | 746637 (43) | 1.03 (1.02-1.04) | 1.0E-07 | CAMK4 | Immune response, inflammation & memory consolidation |
| rs4706020 | 5:130674076 | A/G (0.34) | 0.95 (0.93-0.96) | 1.12E-11 | 518425 (35) | 2.7E-11 | 527801 (39) | 0.98 (0.98-0.99) | 6.4E-09 | CDC42SE2 | F-actin accumulation at immunological synapse of T cells |
| rs4705908 | 5:131347520 | A/G (0.37) | 0.95 (0.93-0.96) | 6.80E-13 | 520344 (36) | 1.6E-11 | 529720 (40) | 0.98 (0.97-0.98) | 8.0E-15 | SLC22A5 | Organic cation transport |
| rs20541 | 5:131995964 | G/A (0.78) | 0.91 (0.89-0.92) | 1.00E-36 | 859747 (38) | 8.4E-51 | 1076820 (42) | 0.92 (0.91-0.92) | 1.2E-129 | SLC22A5 | Organic cation transport |
| rs1145503346 | 5:172192350 | T/C (0.04) | 0.89 (0.86-0.92) | 3.62E-11 | 855569 (33) | 1.3E-10 | 862049 (35) | 0.94 (0.93-0.95) | 3.2E-17 | ERGIC1 | Transport between endoplasmic reticulum and golgi |
| rs41293876 | 6:31466536 | C/G (0.14) | 0.90 (0.88-0.93) | 7.02E-16 | 645820 (36) | 6.5E-18 | 865966 (40) | 0.95 (0.95-0.96) | 4.3E-32 | TNF | Cytokine signalling in immune system |
| rs12153855 | 6:32074804 | C/T (0.10) | 0.92 (0.90-0.94) | 1.96E-11 | 812536 (37) | 2.8E-10 | 821912 (41) | 0.96 (0.95-0.97) | 2.3E-18 | ATF6B | Endoplasmic reticulum stress response |
| rs28383330 | 6:32600340 | G/A (0.13) | 0.88 (0.85-0.90) | 1.42E-18 | 625716 (28) | 1.8E-17 | 632956 (31) | 0.94 (0.93-0.95) | 2.4E-51 | AGER | Immunoglobulin surface receptor |
| rs9275218 | 6:32658933 | G/C (0.34) | 1.06 (1.04-1.08) | 5.36E-10 | 505320 (34) | 1.0E-09 | 512560 (37) | 1.01 (1.01-1.02) | 1.0E-04 | HLA-DRA | Immune response antigen presentation |
| rs629326 | 6:159496713 | T/G (0.61) | 0.95 (0.94-0.97) | 1.7E-12 | 859747 (38) | 4.5E-12 | 1081160 (43) | 0.95 (0.95-0.96) | 5.4E-61[a] | TAGAP[b] | T cell activation |
| rs952558 | 8:81288734 | T/A (0.62) | 0.94 (0.93-0.95) | 3.60E-20 | 862259 (39) | 1.3E-19 | 1083670 (44) | 0.97 (0.96-0.97) | 2.2E-31 | ZBTB10 | Transcriptional regulation |
| rs6996614 | 8:126609868 | A/C (0.53) | 1.07 (1.05-1.08) | 8.48E-17 | 693031 (37) | 1.0E-17 | 914443 (42) | 1.03 (1.02-1.03) | 1.5E-19 | TRIB1 | Protein kinase regulation |
| rs12251307 | 10:6123495 | T/C (0.12) | 1.10 (1.08-1.12) | 1.98E-20 | 864982 (40) | 8.4E-19 | 1086390 (45) | 1.10 (1.09-1.11) | 4.7E-107 | IL2RA | Cytokine signalling in immune system |
| rs10796303 | 10:6627700 | C/T (0.66) | 0.96 (0.94-0.97) | 8.69E-10 | 856884 (38) | 8.5E-10 | 1078300 (43) | 0.97 (0.96-0.97) | 5.6E-25 | PRKCQ | T cell activation |
| rs10822037 | 10:64376558 | C/T (0.61) | 1.06 (1.05-1.08) | 8.53E-19 | 864982 (40) | 1.3E-24 | 1086390 (45) | 1.05 (1.04-1.05) | 4.0E-55 | ADO | Taurine biosynthesis |
| rs10836538 | 11:36365253 | T/G (0.34) | 0.96 (0.94-0.97) | 9.18E-11 | 863063 (39) | 1.1E-13 | 1084480 (44) | 0.95 (0.95-0.96) | 6.2E-55 | PRR5L | Protein phosphorylation |
| rs28520436 | 11:36428447 | T/C (0.03) | 1.20 (1.16-1.24) | 1.22E-24 | 855865 (29) | 4.1E-25 | 1074380 (32) | 1.18 (1.16-1.20) | 5.3E-81 | PRR5L | Protein phosphorylation |
| rs10791824 | 11:65559266 | G/A (0.58) | 1.10 (1.08-1.11) | 1.34E-43 | 864982 (40) | 1.2E-51 | 1086390 (45) | 1.07 (1.06-1.07) | 1.2E-105 | MAP3K11 | Cytokine signalling in immune system |
| rs7936323 | 11:76293758 | A/G (0.46) | 1.08 (1.07-1.10) | 2.0E-34 | 864982 (40) | 1.8E-39 | 1086390 (45) | 1.07 (1.07-1.08) | 1.9E-133 | LRRC32 | TGF beta regulation incl. on T cells |

**Table 1 (continued)**

| Variant | Chr:position | Alleles (EAF) | European discovery | | | Multi-ancestry discovery | | 23andMe European replication (N = 2,904,664) | | | Gene | Pathway/Function |
|---|---|---|---|---|---|---|---|---|---|---|---|---|
| | | | OR (CI) | P | N (studies) | P | N (studies) | OR (CI) | P | | | |
| rs11236813 | 11:76343427 | C/G (0.10) | 0.93 (0.91–0.95) | 1.94E-12 | 864646 (39) | 4.8E-12 | 1086060 (44) | 0.95 (0.94–0.96) | 2.6E-26 | LRRC32 | TGF beta regulation incl. on T cells |
| rs10790275 | 11:118745884 | C/G (0.80) | 1.06 (1.04–1.07) | 5.46E-11 | 859747 (38) | 4.8E-09 | 1081160 (43) | 1.02 (1.02–1.03) | 1.0E-10 | DDX6[b] | mRNA degradation |
| rs7127307 | 11:128187383 | C/T (0.49) | 0.95 (0.93–0.96) | 1.29E-16 | 859747 (38) | 1.0E-17 | 1081160 (43) | 0.96 (0.95–0.96) | 6.1E-52 | FLI1 | NF-kappaB signalling |
| rs705699 | 12:56384804 | A/G (0.40) | 1.04 (1.03–1.05) | 3.31E-09 | 864982 (40) | 6.7E-08 | 1086390 (45) | 1.03 (1.03–1.04) | 8.7E-27 | RPS26 | Peptide chain elongation |
| rs2227491 | 12:68646521 | C/T (0.61) | 1.05 (1.04–1.07) | 1.46E-15 | 864982 (40) | 1.9E-15 | 1086390 (45) | 1.05 (1.05–1.06) | 1.2E-71 | IL22 | Cytokine signalling in immune system |
| rs2415269 | 14:35638937 | A/G (0.26) | 0.94 (0.93–0.96) | 2.26E-16 | 862613 (39) | 9.3E-15 | 1084020 (44) | 0.96 (0.96–0.97) | 3.8E-32 | SRP54 | Peptide chain elongation |
| rs4906263 | 14:103249127 | C/G (0.65) | 1.06 (1.04–1.07) | 2.65E-12 | 693031 (37) | 1.5E-10 | 702407 (41) | 1.04 (1.03–1.04) | 2.9E-36 | TRAF3 | Cytokine signalling in immune system |
| rs2041733 | 16:11229589 | C/T (0.54) | 0.92 (0.91–0.93) | 7.85E-36 | 864982 (40) | 5.8E-40 | 1086390 (45) | 0.94 (0.94–0.95) | 4.2E-95 | RMI2 | DNA repair |
| rs1358175 | 17:38757789 | T/C (0.63) | 1.05 (1.03–1.06) | 1.99E-11 | 864982 (40) | 1.4E-14 | 1086390 (45) | 1.03 (1.03–1.04) | 1.2E-26 | CCR7 | B and T lymphocyte activation |
| rs17881320 | 17:40485239 | T/G (0.08) | 1.09 (1.07–1.12) | 5.34E-13 | 862032 (38) | 2.0E-11 | 870142 (41) | 1.07 (1.06–1.08) | 9.8E-39 | STAT3[b] | Cytokine signalling in immune system |
| rs4247364 | 17:43336687 | C/G (0.70) | 0.96 (0.95–0.98) | 4.54E-08 | 862470 (39) | 1.3E-07 | 1083880 (44) | 0.97 (0.97–0.98) | 1.7E-17 | DCAKD[b] | Coenzyme A biosynthetic process |
| rs56308324 | 17:45819206 | T/A (0.13) | 1.06 (1.04–1.08) | 4.89E-10 | 860694 (38) | 1.1E-08 | 1082110 (43) | 1.03 (1.02–1.04) | 2.6E-11 | TBX21[b] | Th1 differentiation |
| rs28406364 | 17:47454507 | T/C (0.38) | 1.06 (1.05–1.07) | 5.01E-18 | 864982 (40) | 2.3E-18 | 1086390 (45) | 1.04 (1.03–1.04) | 1.5E-34 | GNGT2 | G protein signalling |
| rs2967677 | 19:8789721 | T/C (0.15) | 1.08 (1.07–1.10) | 3.35E-20 | 861624 (38) | 5.8E-23 | 1083040 (43) | 1.06 (1.05–1.07) | 7.5E-49 | CERS4 | Sphingolipid metabolism |
| rs6062486 | 20:62302539 | A/G (0.69) | 1.09 (1.07–1.10) | 5.03E-30 | 782263 (37) | 4.4E-32 | 1003680 (42) | 1.07 (1.07–1.08) | 4.5E-109 | RTEL1 | DNA repair |
| rs4821569 | 22:37316873 | G/A (0.53) | 1.05 (1.04–1.06) | 3.14E-13 | 863063 (39) | 1.6E-11 | 1084480 (44) | 1.04 (1.04–1.05) | 5.4E-50 | CSF2RB | Cytokine signalling in immune system |

The lead SNP at each independent locus is displayed, along with the results from the European-only discovery, multi-ancestry discovery and European replication. The top ranked gene from our gene prioritisation is listed, along with a description of the pathway/function of the gene. The evidence implicating each gene is presented in Supplementary Data 11.

Alleles are listed as effect allele/other allele, the effect allele frequency (EAF) in Europeans (average EAF, weighted by the sample size of each cohort).

Association statistics, Odds ratios (with 95% confidence intervals) and (unadjusted, two-sided) P-values are displayed for the fixed effects European-only meta-analysis and the replication analysis. P-values (unadjusted, two-sided) only are available from the MR-MEGA meta-regression multi-ancestry analysis.

Genome build = GRCh37/hg19.

[a]Imputation batch effect observed in 23andMe data.

[b]One of two or three tied genes at these loci are shown.

**Table 2 | Novel genome-wide significant loci in European-only analysis**

| Variant | Chr:position | Alleles (EAF) | European Discovery | | | Multi-ancestry discovery | | 23andMe European replication (N = 2,904,664) | | Gene | Pathway |
|---|---|---|---|---|---|---|---|---|---|---|---|
| | | | OR (CI) | P | N (studies) | P | N (studies) | OR (CI) | P | | |
| rs301804[b] | 1:8476441 | G/C (0.30) | 1.05 (1.03-1.07) | 2.3E-09 | 698266 (39) | 8.5E-09 | 707642 (43) | 1.03 (1.02-1.03) | 5.5E-16 | RERE | Apoptosis |
| rs61776548 | 1:12091024 | A/G (0.47) | 1.04 (1.02-1.05) | 4.2E-08 | 787144 (39) | 1.4E-07 | 1008560 (44) | 1.02 (1.01-1.02) | 5.6E-09 | TNFRSF1B | Cytokine signalling in immune response |
| rs12565349 | 1:110371629 | G/C (0.15) | 1.05 (1.03-1.07) | 1.3E-08 | 862259 (39) | 1.9E-07 | 1083670 (44) | 1.03 (1.02-1.04) | 5.8E-15 | CSF1 | Cytokine signalling in immune response |
| rs187080438 | 1:150374354 | T/C (0.03) | 1.17 (1.11-1.23) | 3.7E-10 | 758729 (20) | 2.2E-12 | 765209 (22) | 1.14 (1.12-1.16) | 2.0E-41 | CTSS | Antigen presentation in immune response |
| rs146527530[b] | 1:151059196 | G/T (0.02) | 1.27 (1.20-1.35) | 5.5E-15 | 744128 (13) | 7.4E-19 | 744128 (13) | 1.25 (1.22-1.28) | 1.5E-88 | CTSS | Antigen presentation in immune response |
| rs115161931[b] | 1:151063299 | T/C (0.04) | 1.18 (1.13-1.23) | 1.0E-13 | 472565 (26) | 3.2E-12 | 479045 (28) | 1.09 (1.08-1.11) | 2.0E-32 | CTSS | Antigen presentation in immune response |
| rs71625130[b] | 1:151625094 | A/G (0.04) | 1.23 (1.18-1.28) | 2.4E-27 | 770827 (25) | 7.2E-30 | 772963 (26) | 1.17 (1.16-1.19) | 1.7E-89 | RORC[c] | Cytokine signalling in immune response |
| rs149199808[b] | 1:151626396 | T/C (0.03) | 1.32 (1.26-1.38) | 4.4E-30 | 756174 (19) | 8.7E-34 | 762654 (21) | 1.24 (1.22-1.26) | 3.1E-134 | RORC | Cytokine signalling in immune response |
| rs821429[b] | 1:153275443 | A/G (0.96) | 0.86 (0.84-0.89) | 5.9E-18 | 852224 (30) | 8.2E-16 | 858704 (32) | 0.91 (0.89-0.92) | 2.7E-38 | S100A7 | Differentiation regulation incl. in the innate immune system |
| rs12138773 | 1:153843489 | A/C (0.03) | 1.11 (1.07-1.16) | 2.3E-08 | 851937 (28) | 1.3E-09 | 858417 (30) | 1.07 (1.05-1.09) | 3.5E-16 | S100A12[c] | Regulation of inflammatory processes and immune response |
| rs67766926[a,b] | 2:61163581 | G/C (0.23) | 1.05 (1.03-1.06) | 5.7E-10 | 863063 (39) | 2.9E-11 | 1084480 (44) | 1.05 (1.04-1.05) | 1.2E-41 | AHSA2P | Protein folding |
| rs112385344 | 2:112275538 | T/C (0.12) | 1.06 (1.04-1.08) | 2.8E-09 | 852837 (34) | 3.9E-08 | 862213 (38) | 1.04 (1.03-1.05) | 1.5E-18 | MERTK[c] | Inhibits TLR-mediated innate immune response |
| rs62193132 | 2:242788256 | T/C (0.46) | 1.04 (1.03-1.06) | 1.5E-09 | 832761 (26) | 7.1E-08 | 1052040 (30) | 1.03 (1.02-1.03) | 1.5E-19 | NEU4 | Sphingolipid metabolism |
| rs10833[b] | 4:142654547 | C/T (0.65) | 1.04 (1.03-1.06) | 7.3E-09 | 859747 (38) | 6.0E-08 | 1081160 (43) | 1.02 (1.02-1.03) | 3.4E-15 | IL15 | Cytokine signalling in immune response |
| rs148161264[b] | 5:14604521 | G/C (0.04) | 1.10 (1.07-1.14) | 7.4E-10 | 850619 (29) | 2.0E-08 | 857099 (31) | 1.05 (1.03-1.06) | 1.6E-08 | OTULINL | Endoplasmic reticulum component |
| rs7701967 | 5:130059750 | A/G (0.31) | 0.95 (0.94-0.97) | 3.4E-09 | 520344 (36) | 3.6E-09 | 529720 (40) | 0.99 (0.98-0.99) | 1.1E-06 | LYRM7 | Mitochondrial respiratory chain complex assembly |
| rs4532376[b] | 5:176774403 | A/G (0.30) | 1.04 (1.03-1.06) | 3.5E-09 | 859747 (38) | 2.3E-09 | 1081160 (43) | 1.03 (1.02-1.03) | 1.4E-18 | RGS14 | G-alpha signalling |
| rs72925996[b] | 6:90930513 | C/T (0.33) | 0.96 (0.94-0.97) | 3.2E-10 | 862259 (39) | 5.4E-09 | 1083670 (44) | 0.96 (0.95-0.96) | 2.2E-44 | BACH2 | NF-kappaB proinflammatory signalling |
| rs989437 | 7:28830498 | G/A (0.61) | 0.96 (0.95-0.97) | 6.1E-11 | 864982 (40) | 1.0E-09 | 1086390 (45) | 0.97 (0.96-0.97) | 6.9E-31 | CREB5[c] | AMPK & ATK signalling |
| rs34215892 | 8:21767240 | A/G (0.03) | 0.87 (0.83-0.90) | 4.7E-11 | 436369 (24) | 2.0E-09 | 442849 (26) | 0.89 (0.88-0.91) | 1.0E-36 | DOK2 | Immune response IL-23 signalling |
| rs118162691 | 8:21767809 | A/C (0.05) | 0.92 (0.89-0.94) | 7.8E-09 | 856229 (30) | 1.8E-07 | 862709 (32) | 0.90 (0.88-0.91) | 1.1E-44 | DOK2 | Immune response IL-23 signalling |
| rs7843258 | 8:141606042 | C/T (0.82) | 1.05 (1.04-1.07) | 1.5E-09 | 859747 (38) | 3.6E-10 | 1081160 (43) | 1.04 (1.03-1.05) | 7.0E-25 | AGO2 | siRNA-mediated gene silencing |
| rs7857407 | 9:33430707 | A/T (0.40) | 1.04 (1.02-1.05) | 2.5E-08 | 864982 (40) | 9.0E-09 | 1086390 (45) | 1.03 (1.02-1.03) | 5.1E-18 | AQP3 | Aquaporin-mediated transport |
| rs10988863 | 9:102331281 | C/A (0.21) | 0.95 (0.93-0.96) | 5.1E-11 | 862259 (39) | 3.0E-09 | 1083670 (44) | 0.97 (0.97-0.98) | 1.3E-13 | NR4A3 | Transcriptional activator |
| rs17368814 | 11:102748695 | G/A (0.13) | 0.95 (0.93-0.97) | 1.4E-08 | 858117 (37) | 6.8E-07 | 1078260 (41) | 0.95 (0.95-0.96) | 1.2E-27 | MMP12 | Extracellular matrix organisation |
| rs11216206 | 11:116843425 | G/C (0.07) | 1.10 (1.07-1.14) | 5.5E-10 | 557183 (35) | 2.9E-10 | 778595 (40) | 1.04 (1.03-1.05) | 8.5E-15 | SIK3 | LKB1 signalling |
| rs5005507[b] | 12:94611908 | C/G (0.74) | 1.05 (1.03-1.06) | 3.6E-09 | 859747 (38) | 9.6E-08 | 1081160 (43) | 1.03 (1.02-1.04) | 2.7E-18 | PLXNC1 | Semaphorin interactions incl. in immune response |
| rs7147439 | 14:105523663 | A/G (0.73) | 0.96 (0.95-0.97) | 4.7E-08 | 781909 (37) | 6.6E-07 | 1003220 (42) | 0.97 (0.96-0.97) | 4.8E-24 | GPR132 | GPCR signalling |
| rs2542147 | 18:12775851 | T/G (0.84) | 0.95 (0.93-0.96) | 1.5E-09 | 862470 (39) | 7.5E-08 | 1083880 (44) | 0.96 (0.95-0.97) | 2.6E-26 | PTPN2 | Cytokine signalling in immune response |

The lead SNP at each independent locus is displayed, along with the results from the European-only discovery, multi-ancestry discovery and European replication. The top-ranked gene from our gene prioritisation is listed, along with a description of the pathway/function of the gene. The evidence implicating each gene is presented in Supplementary Data 11.

Alleles are listed as Effect allele/other allele, the effect allele frequency (EAF) in Europeans (average EAF, weighted by the sample size of each cohort).

Genome build = GRCh37/hg19.

[a] rs4643526 at the same locus was previously identified in the discovery analysis of Paternoster et al.[2]. However, this association did not replicate in that study.

[b] Whilst not identified in any GWAS AD papers, these loci have previously shown evidence for association with AD in supplementary material of methodological papers[92,93].

[c] One of two or three tied genes are shown.

**Table 3 | Additional loci associated with the multi-ancestry analysis**

| Variant | Chr:position | Alleles (EAF) | Multi-ancestry discovery N = 992,907 P | European discovery N = 864,982 P | RIKEN - Bio-bank Japan N = 118,287 P | 23andMe Latino N = 525,348 P | 23andMe African N = 174,015 P | 23andMe European N = 2,904,664 P | Known Associations | Novel Associations |
|---|---|---|---|---|---|---|---|---|---|---|
| rs114059822[a] | 1:19804918 | T/G (0.03) | 8.59E-09 | 0.25 | – | 0.07 | 0.03 | 0.87 | NA | NA |
| rs9247 | 2:234113301 | T/C (0.21) | 1.92E-09 | 7.32E-08 | 7.71E-05 | 1.49E-13 | 7.23E-03 | 2.93E-51 | | **all[b]** |
| rs9864845 | 3:112383847 | A/G (0.37) | 2.17E-12 | 0.22 | 3.92E-13 | 0.75 | 0.23 | 0.12 | Japanese (Tanaka et al.[8]) | |
| rs34599047 | 6:106629690 | C/T (0.18) | 3.32E-08 | 1.29E-07 | 0.03 | 7.18E-04 | 0.02 | 3.23E-22 | | **all[b]** |
| rs7773987 | 6:135707486 | T/C (0.60) | 1.22E-08 | 9.57E-08 | 0.15 | 0.18 | 1.95E-03 | 5.93E-13 | | **European, African** |
| rs118029610[a] | 9:1894613 | T/C (0.03) | 1.89E-08 | 2.97E-04 | – | 0.5 | 0.31 | 0.78 | NA | NA |
| rs117137535 | 9:140500443 | A/G (0.03) | 1.99E-08 | 5.50E-08 | – | 3.99E-07 | 0.33 | 9.25E-19 | European (Grosche et al.[7]) | Latino |
| rs4312054 | 11:7977161 | G/T (0.43) | 3.21E-12 | 0.86 | 3.46E-15 | 0.4 | 0.33 | 0.52 | Japanese (Tanaka et al.[8]) | |
| rs150113720[a] | 11:83439186 | G/C (0.02) | 5.52E-10 | 0.40 | – | 0.1 | 0.22 | 0.14 | NA | NA |
| rs115148078[a] | 11:101361300 | T/C (0.02) | 5.91E-09 | 0.37 | – | 3.69E-03 | 0.91 | 0.89 | NA | NA |
| rs4262739 | 11:128421175 | A/G (0.50) | 2.20E-08 | 6.03E-07 | 2.28E-03 | 1.89E-06 | 0.09 | 1.45E-36 | European & Japanese (Tanaka et al.[8]) | Latino |
| rs1059513 | 12:57489709 | C/T (0.08) | 5.15E-09 | 1.57E-07 | 0.33 | 3.06E-04 | 0.17 | 6.95E-16 | European (Tanaka et al.[8]) | Latino |
| rs4574025 | 18:60009814 | T/C (0.55) | 7.00E-10 | 1.48E-06 | 2.67E-05 | 2.59E-04 | 1.24E-05 | 2.96E-05 | European & Japanese (Tanaka et al.[8]) | Latino, African |
| rs6023002 | 20:52797237 | C/G (0.52) | 4.05E-10 | 2.26E-06 | 2.82E-07 | 5.96E-03 | 0.07 | 3.22E-28 | European & Japanese (Tanaka et al.[8]) | Latino |

For loci that were associated in the multi-ancestry discovery analysis, but not the European discovery analysis, we show the (unadjusted two-sided) P-values for association across 4 diverse ancestral groups, European, Japanese, Latino and African. Full association statistics (including OR and 95% CI) for each variant can be viewed in Supplementary Data 4 (and results across all cohorts individually are depicted in Supplementary Fig. 2).

Alleles are reported as effect allele/other allele.

Genome build = GRCh37/hg19.

NA indicates finding not replicated and likely to be false-positive in discovery.

Bold is used in the novel column to denote the 3 associations that are entirely novel (i.e. locus has not been associated in any ancestry previously).

– Variant was not available in dataset.

[a]Genome-wide significant loci without replication that are assumed to be false positives in the discovery data.

[b]Whilst not identified in any GWAS AD papers, these loci have previously shown evidence for association with AD in the supplementary material of methodological papers[92] or GWAS of combined allergic disease phenotype[5].

the replication samples (of European, Latino and/or African ancestry; Bonferroni corrected $P < 0.05/14 = 3.6 \times 10^{-3}$). Two index SNPs which did not replicate in any of the samples (rs9864845 (near *CCDC80*), rs4312054 (near *NLRP10*)) appear to have been driven by association in the Japanese RIKEN study only (Supplementary Data 4, Supplementary Figs. 2, 3). Whilst the allele frequencies of these index SNPs are similar between Europeans and Japanese (37% vs 42% for rs9864845, 41% vs 46% for rs4312054, Supplementary Data 5), in a multi-ancestry fixed effect meta-analysis at both these loci there were neighbouring (previously reported)[8] SNPs with stronger evidence of association (rs72943976, $P = 2 \times 10^{-9}$ and rs59039403 $P = 2 \times 10^{-35}$, Supplementary Fig. 3), that did show large allele frequencies for Japanese (~34% and 13%, respectively) but <1% in Europeans. A further 4 loci did not replicate, and on closer examination (Supplementary Fig. 2, and MAF in cases <1%), their association in the discovery analysis appeared to be driven by a false positive outlying result in a single European cohort.

Seven of the loci in Table 3 have been previously reported as associated with AD. Two (rs117137535 (near *ARRDC1*)[7] and rs1059513 (near *STAT6*)[8]) were previously only associated with Europeans (and these were variants that were just below the genome-wide significance threshold in our European only analysis). Three (rs4262739 (near *ETS1*), rs4574025 (within *TNFRSF11A*) and rs6023002 (near *CYP24A1*)) were previously associated in Japanese and Europeans[8], while 2 were previously associated only in Japanese[8,10], using the same Japanese data (RIKEN) that we include here. Therefore, in our multi-ancestry analysis (and replication) we identify 3 loci that have not previously been reported in a GWAS of AD of any ancestry (rs9247 (near *INPP5D*), rs34599047 (near *ATG5*) and rs7773987 (near *AHI1*)), all of which are associated in two or more populations in our data (Table 3).

In addition, for 5 loci which had previously been associated with individuals of European and/or Japanese ancestry, we now show evidence that these are also associated with individuals of Latino ancestry and one is also associated in individuals of African ancestry (Table 3).

## Comparison of associations between ancestries

Effect sizes of the index SNPs were remarkably similar between individuals of European and Latino ancestry (Supplementary Fig. 4A). There were only two variants with any evidence for a difference (where Latino $P > 5 \times 10^{-4}$ and the 95% confidence intervals didn't overlap), but the plot shows that these were only marginally different and likely to be due to chance. Effect size comparison of the index SNPs between individuals of European and African ancestry showed greater differences (Supplementary Fig. 4B). 17 SNPs showed some evidence for being European-specific in that comparison. The confidence intervals in the Japanese data were much wider but there was weak evidence for one SNP being European-specific and stronger evidence for two SNPs being Japanese-specific (Supplementary Fig. 4C). These were rs4312054 (JAP CI: 0.75-0.84, EUR CI: 0.99-1.01) and rs9864845 (JAP CI: 1.16-1.30, EUR CI: 0.99-1.06), mentioned earlier as the SNPs that appeared to be driven only by Japanese individuals in the multi-ancestry meta-analysis (Supplementary Data 4).

## Established associations

A review of previous work in this field (Supplementary Data 1) shows that a total of 202 unique variants (across a much smaller number of loci) have been reported to be associated with AD. We found evidence for all but 7 variants of these being nominally associated in the current GWAS (81% in the European and 96% in the multi-ancestry analysis). Variants we did not find to be associated were either rare variants (MAF < 0.01), or insertion/deletion mutations, which were not included in our analysis.

## Genetic correlation between AD and other traits

LD score regression analyses showed high genetic correlation, as expected, between AD and related allergic traits, e.g. asthma (rg=0.53, $P = 2 \times 10^{-32}$), hay fever (rg=0.51, $P = 7 \times 10^{-17}$) and eosinophil count (rg = 0.27, $P = 1 \times 10^{-7}$) (Supplementary Fig. 5 and Supplementary Data 6). In addition, depression and anxiety showed notable

genetic correlation with AD (rg = 0.17, $P = 2 \times 10^{-7}$), a relationship which has been reported previously, but causality has not been established[17]. Furthermore, gastritis also showed substantial genetic correlation (rg = 0.31, $P = 1 \times 10^{-5}$), which may be due to the AD genetic signal including variants with pervasive inflammatory function or the observed correlation could indicate a shared risk locus for inflammation or microbiome alteration in the upper gastrointestinal tract, or it may reflect the use of systemic corticosteroid treatment for atopic disease which in some cases causes gastritis as a side effect.

## Tissue, cell and gene-set enrichment

The tissue enrichment analyses using distinct molecular evidence (representing open chromatin and gene expression) both found blood to be the tissue showing strongest enrichment of GWAS loci (Fig. 2). The Garfield test for enrichment of genome-wide loci (with $P < 1 \times 10^{-8}$) in DNase I hypersensitive sites (DHS broad peaks) found evidence of enrichment ($P < 0.00012$) in 41 blood tissue analyses, a greater signal than another tissue or cell type (Fig. 2a and Supplementary Data 7). The strongest enrichment (OR > 5.5 and $P < 1 \times 10^{-10}$) was seen for T-cell, B-cell and natural killer lymphocytes (CD3+, CD4+, CD56+ and CD19+). As expected for AD, Th2 showed stronger enrichment (OR = 4.3, $P = 1 \times 10^{-8}$) than Th1 (OR = 2.3, $P = 2 \times 10^{-4}$). The strongest enrichment in tissue samples representing skin was seen for foreskin keratinocytes (OR = 2.0, $P = 0.008$), but this did not meet a Bonferroni-corrected $P$-value threshold ($0.05/425 = 1 \times 10^{-4}$).

The most enriched tissue type in MAGMA gene expression enrichment analysis was whole blood ($P = 2 \times 10^{-14}$). Others that met our Bonferroni-corrected $P$-value ($P < 0.0009$) were spleen, EBV-transformed lymphocytes, sun-exposed and unexposed skin, small intestine and lung (Fig. 2b and Supplementary Data 8).

DEPICT cell-type enrichment analysis identified a similar set of enriched cell-types: blood, leucocytes, lymphocytes and natural killer cells, but with the addition that the strongest enrichment was seen for synovial fluid ($P = 2 \times 10^{-7}$), which may be due to its immune cell component.

The DEPICT pathway analysis found 420 GO terms with enrichment (FDR < 5%) amongst the genes from our GWAS loci (Supplementary Data 9). The pathway with the strongest evidence of enrichment was 'hemopoietic or lymphoid organ development' ($P = 1 \times 10^{-16}$). All terms with FDR < 5% are represented in Supplementary Fig. 6, where the terms are grouped according to similarity and the parent terms labelled illustrating the strong theme of immune system development and signalling.

## Gene prioritisation and biological interpretation in silico

The top genes prioritised using our composite score from publicly available data for each of the established European AD loci are shown in Table 1 and Fig. 3a (and the evidence that makes up the prioritisation scores is shown in Supplementary Fig. 7). The top three prioritised genes at each independent locus are shown in Supplementary Data 10 and a summary of all evidence for all genes reviewed in silico is presented in Supplementary Data 11.

In most cases the top prioritised gene had been implicated (in previous GWAS) or is only superseded marginally by an alternative candidate. One interesting exception is on chromosome 11, where *MAP3K11* (with a role in cytokine signalling – regulating the JNK signalling pathway) is markedly prioritised over the previously implicated *OVOL1*[18] (involved in hair formation and spermatogenesis), although the prioritisation of *MAP3K11* is predominantly driven by TWAS evidence in multiple cell types rather than colocalisation or other evidence.

There are three instances where multiple associations in the region implicate additional novel genes. Two are genes involved in TLR4 signalling: *S100A9* (prioritised in addition to the established *FLG*

and *IL6R* on chromosome 1) and *AGER* (prioritised in addition to *HLA-DRA* on chromosome 6). The third has a likely role in T-cell activation: *CDC42SE2* (prioritised in addition to *SLC22A5* on chromosome 5).

The top prioritised gene at each of the novel European loci are shown in Table 2 and Fig. 3b. Many are in pathways already identified by previous findings (e.g. cytokine signalling—specially IL-23, antigen presentation and NF-kappaB proinflammatory response). At one locus, the index SNP, rs34215892 is a missense (Pro274Leu) mutation within the *DOK2* gene, although this mutation is categorised as tolerated or benign by SIFT and PolyPhen. The genes with the highest prioritisation score amongst the novel loci were *GPR132* (total evidence Score=24), *NEU4* (score=22), *TNFRSF1B* (score = 19) and *RGS14* (score=19) and each show biological plausibility as candidates for AD pathogenesis.

*GPR132* is a proton-sensing transmembrane receptor, involved in modulating several downstream biological processes, including immune regulation and inflammatory response, as reported previously in an investigation of this protein's role in inflammatory bowel disease[19]. The index SNP at this locus, rs7147439 (which was associated with Europeans, Latinos, Africans, but not Japanese), is an intronic variant within the *GPR132* gene. The AD GWAS association at this locus colocalises with the eQTL association for *GPR132* in several immune cell types (macrophages[20], neutrophils[21], several T-cell datasets[22]) as well as in colon, lung and small intestine in GTEx[23]. *GPR132* has also been shown to be upregulated in lesional and nonlesional skin in AD patients, compared to skin from control individuals[24,25]. OpenTargets and POSTGAP both prioritise *GPR132* for this locus.

The SNP rs62193132 (which showed consistent effects in European, Latino and Japanese individuals, but little evidence for association in African individuals, Supplementary Fig. 2), is in an intergenic region between *NEU4* (-26 kb) and *PDCD1* (-4 kb away) on chromosome 2. *NEU4* was the highest scoring in our gene prioritisation pipeline (score=22). However, *PDCD1* also scores highly (score = 18, Supplementary Data 10). NEU4 is an enzyme that removes sialic acid residues from glycoproteins and glycolipids, whereas PDCD1 is involved in the regulation of T cell function. The AD GWAS association at this locus colocalises with the eQTL for *NEU4* in several monocyte and macrophage datasets[22,26–28] as well as in the ileum, colon and skin[23,29]. The eQTL for *PDCD1* also colocalises in monocytes and macrophages[27,28] as well as T-cells[22], skin and whole blood[23]. In addition to the eQTL evidence, *PCDC1* is upregulated in lesional and non-lesional skin in AD patients compared to skin from control individuals[24,25]. OpenTargets and PoPs prioritise *NEU4*, whilst POSTGAP prioritises *PDCD1* at this locus.

*TNFRSF1B* is part of the TNF receptor, with an established role in cytokine signalling. rs61776548 (which showed consistent associations across all major ancestries tested) is 136 kb upstream of *TNFRSF1B*, actually within an intron of *MIIP*. *MIIP* encodes Migration and Invasion-Inhibitory Protein, which may function as a tumour suppressor. However, *TNFRSF1B* is a stronger candidate gene since the AD GWAS association at this locus colocalises with the eQTL for *TNFRSF1B* T cells[22,30], macrophages[20], fibrobasts[31] and platelets[29]. Furthermore, *TNFRSF1B* gene expression and the corresponding protein are upregulated in lesional and nonlesional skin compared to controls[24,25,32] and the PoPs method prioritised this gene at this locus.

*RGS14* is a multifunctional cytoplasmic-nuclear shuttling protein which regulates G-protein signalling, but whose role in the immune system is yet to be established. rs4532376 is 10.5 kb upstream of *RGS14* and within an intron of *LMAN2*. The AD GWAS association at this locus colocalises with the eQTL for *RGS14* in macrophages[20], CD8 T-cells[22], blood[33] and colon[23]. *RGS14* has also been shown to be upregulated in lesional skin of AD cases compared to skin from control individuals[25] and DEPICT prioritises this gene. However, at this locus *LMAN2* is also a reasonably promising candidate (score=15) based on colocalisation and differential expression evidence (Supplementary Data 11). Open-Targets and POSTGAP prioritise this alternative gene at this locus and

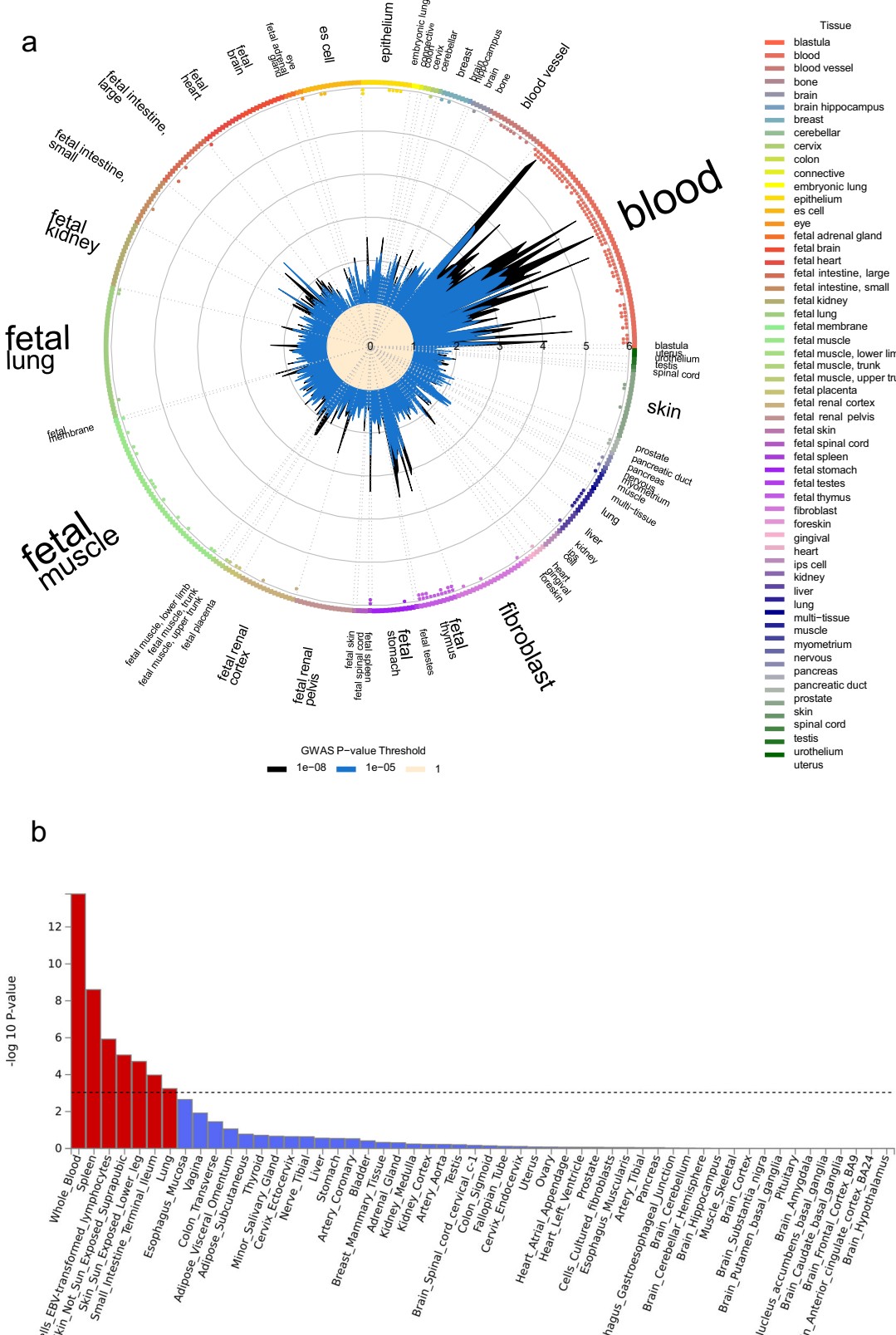

**Fig. 2 | Cell type tissue enrichment analysis. a** GARFIELD enrichment analysis of open chromatin data. Plot shows enrichment for AD associated variants in DNase I Hypersensitive sites (broad peaks) from ENCODE and Roadmap Epigenomics datasets across cell types. Cell types are sorted and labelled by tissue type. ORs for enrichment are shown for variants at GWAS thresholds of $P < 1 \times 10^{-8}$ (black) and $P < 1 \times 10^{-5}$ (blue) after multiple-testing correction for the number of effective annotations. Outer dots represent enrichment thresholds of $P < 1 \times 10^{-5}$ (one dot)

and $P < 1 \times 10^{-6}$ (two dots). Font size of tissue labels corresponds to the number of cell types from that tissue tested. **b** MAGMA enrichment analysis of gene expression data. Plot shows $P$-value for MAGMA enrichment for AD associated variants with gene expression from 54 GTEx ver.8 tissue types. The enrichment $-\log_{10}(P\text{-value})$ for each tissue type is plotted on the $y$-axis. The Bonferroni corrected threshold $P = 0.0009$ is shown as a dotted line and the 7 tissue types that meet this threshold are highlighted as red bars.

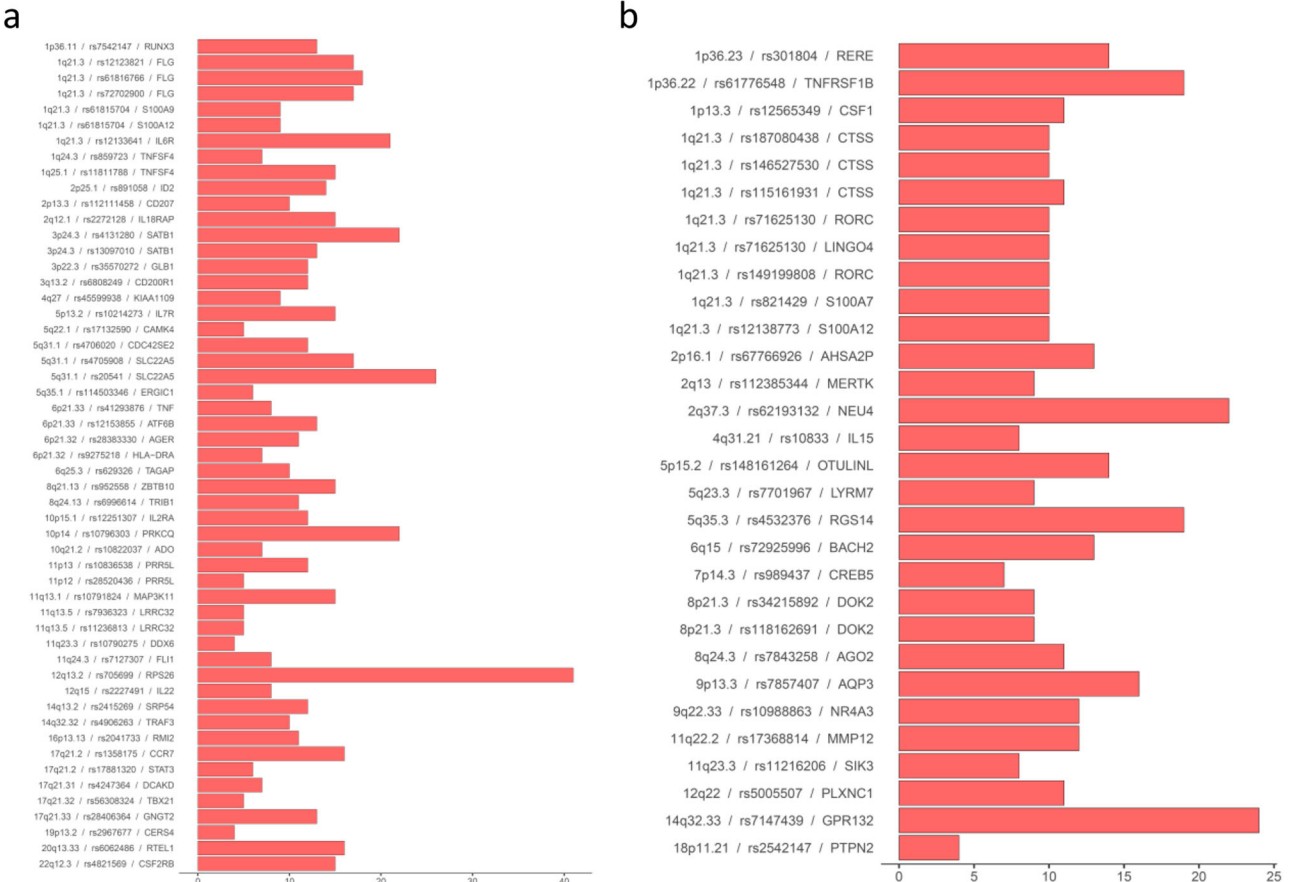

**Fig. 3 | Prioritised genes at GWAS loci.** Prioritised genes at known (**a**) and novel (**b**) loci. For each independent GWAS locus the top prioritised gene (or genes if they were tied) from our bioinformatic analysis is presented along with a bar representing the total evidence score for that gene. A more detailed breakdown of the constituent parts of this evidence score is presented in Supplementary Fig. 5 and the total evidence scores for the top 3 genes at each locus are presented in Supplementary Data 10. NB. There are some cases of two independent GWAS signals implicating the same gene.

it is possible that genetic variants at this locus influence AD risk through both genetic mechanisms.

We did not include the 3 novel variants from the multi-ancestry analysis in the comprehensive gene prioritisation pipeline because the available resources used predominantly represent European samples only. We did however investigate these variants using Open Targets Genetics, to identify any evidence implicating specific genes at these loci. rs9247 is a missense variant in *INPP5D*, encoding SHIP1, a protein that functions as a negative regulator of myeloid cell proliferation and survival. The *INPP5D* gene has been implicated in hay fever and/or eczema[5] and other epithelial barrier disorders including inflammatory bowel disease. rs7773987 is intronic for *AHI1 (*Abelson helper integration site 1) which is involved with brain development but expressed in a range of tissues throughout the body; single cell analysis in skin shows expression in multiple cell types including specialised immune cells and keratinocytes, but the highest abundance is in endothelial cells (data available from v21.1 proteinatlas.org). The closest genes to rs34599047 are *ATG5* (involved in autophagic vesicle formation) and *PRDM1* (which encodes a master regulator of B cells).

### Network analysis

STRING network analysis of the 70 human proteins encoded by genes listed in Tables 1 and 2 showed a protein-protein interaction (PPI) enrichment $p$-value $< 1 \times 10^{-16}$. The five most highly significant (FDR $P = 1 \times 10^{-9}$) Gene Ontology (GO) terms for biological process relate to immune system activation and regulation (Supplementary Data 12). The network described by the highly enriched term 'Regulation of immune system process' (GO:0002682) is shown in Fig. 4.

Extending the network to include the less well characterised genes/proteins from the multi-ancestry analysis further strengthened this predicted network: The PPI enrichment was again $P < 1 \times 10^{-16}$ and 'Regulation of immune system process' was the most enriched term (FDR $P = 5 \times 10^{-13}$).

## Discussion

We present the results of a comprehensive genome-wide association meta-analysis of AD in which we have identified a total of 91 associated loci. This includes 81 loci identified amongst individuals of European ancestry replicated in a further sample of 2.9 million European individuals (as well as many showing replication in data for other ancestries). Of the additional 10 loci identified in a multi-ancestry analysis, 8 replicated in at least one of the populations tested (European, Latino and African ancestry) and a further 2 may be specific to individuals of East Asian ancestry (but require replication).

The majority of the loci associated with AD are shared between the ancestry groups represented in our data, though there were some notable exceptions. We report two previously identified loci with associations that appear to be specific to the Japanese cohort (although driven by just one cohort and still require independent replication). Whilst these have been previously reported[8], this used the same data as examined here. However, rs59039403 within *NLRP10* is a likely deleterious missense mutation at reasonable frequency in Japanese (13%) that is present at a far lower frequency (<1%) in Europeans. Equally, previous further investigation of the association near *CCDC80* found a putative functional variant (rs12637953) that affects the expression of an enhancer (associated with *CCDC80* promoter) in

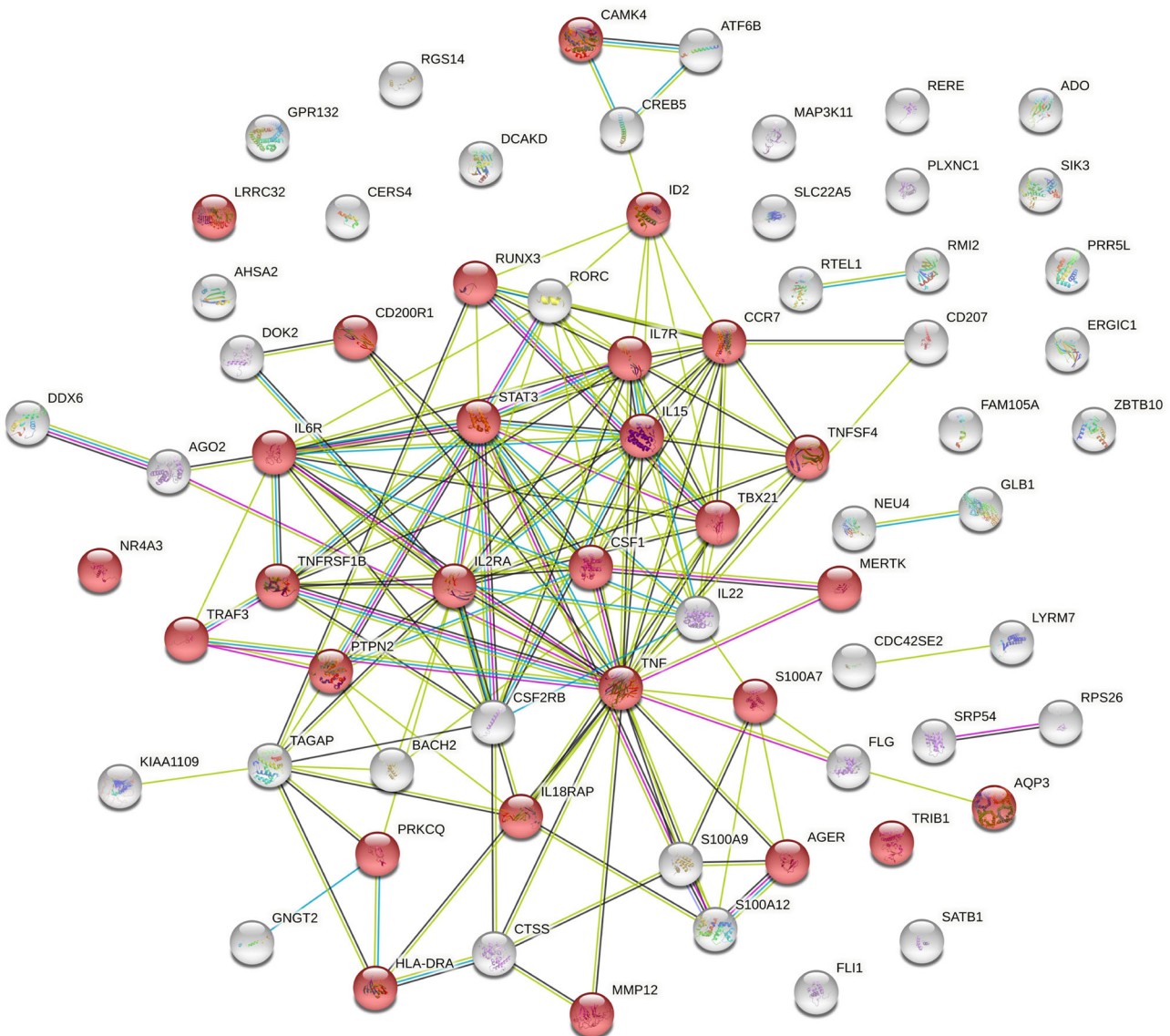

**Fig. 4 | Predicted interaction network of proteins encoded by the top prioritised genes from known and novel European GWAS loci.** Protein-protein interaction analysis carried out in STRING v11.5; nodes coloured red represent the GO term 'Regulation of immune system process' (GO:0002682) for which 28/1514 proteins are included (FDR $P = 1 \times 10^{-9}$). Full results for all identified pathways are available in Supplementary Data 12.

epidermis and Langerhans cells[8], increasing the evidence that these Japanese-specific loci are real. Furthermore, we have identified several loci with association in Europeans (many of which also showed association in individuals of Japanese or Latino ancestry) but which showed no evidence of association in individuals of African ancestry. It is tempting to speculate, using our knowledge of the differing AD phenotypes between European, Asian and African people[34,35] that the differing genetic associations at some loci may contribute to these clinical observations. rs7773987 within an intron of *AHI1* may, for example, indicate a mechanism contributing to neuronal sensitisation leading to the marked lichenification and nodular prurigo-type lesions[36] that characterise AD in some people of African and European ethnicities[37]. Large-scale population cohorts (as used here) have been useful for identifying associated variants. However, we do note that the variants identified should be further examined with respect to specific aspects of AD (age of onset, severity and longitudinal classes[38]) in future analysis.

The dominance of blood as the tissue showing most enrichment of our GWAS signals in regions of DNAse hypersensitivity and of eQTLs suggests the importance of systemic inflammation in AD and this is in

keeping with knowledge of the multisystem comorbidities associated with AD[39]. The dominance of blood also supports the utility of this easily accessible tissue when characterising genetic risk mechanisms, and for the measurement of biomarkers for many of the implicated loci. However, skin tissue also showed enrichment and there are likely to be some genes for which the effect is only seen in skin. For example, we know that two genes previously implicated in AD, *FLG* and *CD207*[2,18] are predominantly expressed in the skin and in our gene prioritisation investigations there was no evidence from blood linking *FLG* to the rs61816766 association and only one analysis of monocytes separated from peripheral blood mononuclear cell (PBMC) samples[28] which implicated *CD207* for the rs112111458 association, amongst an abundance of evidence from skin for both genes playing a role in AD (Supplementary Data 11). So, whilst the enrichment analysis suggests blood as a useful tissue for genome scale studies of AD and a reasonable tissue to include for further investigation at specific loci, it does not preclude skin as the more relevant tissue for a subset of important genes.

At many of the loci identified in this GWAS, our gene prioritisation analysis, as well as the DEPICT pathway analysis, implicated genes from

pathways that are already known to have a role in AD pathology. The overwhelming majority of these are in pathways related to immune system function; STRING network analysis highlighted the importance of immune system regulation, in keeping with an increasing awareness of the importance of balance in opposing immune mechanisms that can cause paradoxical atopic or psoriatic skin inflammation[40]. Whilst our in silico *analyses* cannot definitively identify specific causal genes (rather, we present a prioritised list of all genes at each locus along with the corresponding evidence for individual evaluation), it is of note that for many of the previously known loci (Table 1) our approach identifies genes which have been validated in experimental settings, e.g. *FLG*[41], *TNF*[42] and *IL22*[43]. The individual components of the gene prioritisation analysis have their limitations, particularly the high probability that findings, whilst demonstrating correlation, do not necessarily provide evidence for a causal relationship. This has been particularly highlighted with respect to colocalisation of GWAS and eQTL associations, where high co-regulation can implicate many potentially causal genes[44]. Another limitation is that only cell types (and conditions) that have been studied and made available are included in the in silico analysis, and gaps in the data may prove crucial. However, we believe this broad-reaching review of complementary datasets and methods is a useful initial approach to summarise the available evidence, prioritise genes for follow-up and provide information to inform functional experiments. The best evidence is likely to be produced from triangulation of multiple experiments and/or datasets and we have presented our workflow and findings in a way to allow readers to make their own assessments. Another important limitation of our gene prioritisation, is that we only undertook the comprehensive approach for loci associated in European individuals, given that the majority of datasets used come from (and may only be relevant for) European individuals. Expansion of resources that allow for similarly comprehensive follow-up of GWAS loci in individuals of non-European ancestry are urgently needed[45]. However, we do report some evidence that implicates certain genes at loci from our multi-ancestry analysis, whilst noting that these require further investigation in appropriate samples from representative populations.

Amongst the genes prioritised at the novel loci identified in this study, four are targets of existing drugs (and have the required direction of action consistent with the AD risk allele's direction of effect on the gene expression) as reported by Open Targets[46]: *CSF1* is targeted by a macrophage colony-stimulating factor 1 inhibiting antibody (in phase II trials as cancer therapy but also for the treatment of rheumatoid arthritis and cutaneous lupus); *CTSS* is targeted by a small molecule cathepsin S inhibitor (in phase I-II trials for coeliac disease and Sjogren syndrome); *IL15*, targeted by an anti-IL-15 antibody (in phase II trials for autoimmune conditions including vitiligo and psoriasis); and *MMP12*, targeted by small molecule matrix metalloprotease inhibitors (in phase III studies for breast and lung cancer, plus phase II for cystic fibrosis and COPD)[47]. These may offer valuable drug repurposing opportunities.

We have presented the largest GWAS of AD to date, identifying 91 robustly associated loci, 22 with some evidence of population-specific effects. This represents a significant increase in knowledge of AD genetics compared to previous efforts, taking the number of GWAS hits identified in a single study from 31 to 91 and making available the well-powered summary statistics to enable many future important studies (e.g. Mendelian Randomization to investigate causal relationships). To aid translation we have undertaken comprehensive post-GWAS analyses to prioritise potentially causal genes at each locus, implicating many immune system genes and pathways and identifying potential novel drug targets.

## Methods

Appropriate ethical approval was obtained for all cohorts by their ethics committees as detailed in the Supplementary Methods.

### Phenotype definition

Cases were defined as those who have "ever had atopic dermatitis", according to the best definition for the cohort, where doctor-diagnosed cases were preferred. Controls were defined as those who had never had AD. Further details on the phenotype definitions for the included studies can be found in Supplementary Methods and Supplementary Data 2.

### GWAS analysis and quality control of summary data

We performed genome-wide association analysis (GWAS) for AD case-control status across 40 cohorts including 60,653 AD cases and 804,329 controls of European ancestry. We also included cohorts with individuals of mixed ancestry (Generation R), as well as Japanese (Biobank Japan), African American (SAGE II and SAPPHIRE) and Latino (GALA II) studies, giving a total of 65,107 AD cases and 1,021,287 controls.

Genetic data was imputed separately for each cohort with the majority of European cohorts using the haplotype reference consortium (HRC version r1.1) reference panel[48] (imputed with either the Michigan or Sanger server). 8 European and 2 non-European cohorts instead used the 1000 Genomes Project Phase 1 reference panel for imputation. GWAS was performed separately for each cohort while adjusting for sex and ancestry principal components derived from a genotype matrix (as appropriate in each cohort). Genetic variants were restricted to a MAF > 1% and an imputation quality score > 0.5 unless otherwise specified in the Supplementary Methods. In order to robustly incorporate cohorts with small sample sizes, we applied additional filtering based on the expected minor allele count (EMAC) as previously demonstrated[49]. EMAC combines information on sample size, MAF and imputation quality (2*N*MAF*imputation quality score) and a threshold of >50 EMAC was used to include variants for all cohorts. QQ-plots and Manhattan plots for each cohort were generated and visually inspected as part of the quality control process.

### Meta-analysis

For the discovery phase, meta-analysis of the European cohorts was performed with GWAMA[47] for 12,147,822 variants assuming fixed effects, while the multi-ancestry analysis of all cohorts was conducted in MR-MEGA[50] (which models the heterogeneity in allelic effects that is correlated with ancestry). The latter included only 8,684,278 variants as MR-MEGA excludes variants where the number of contributing cohorts is less than 6. $P < 5 \times 10^{-8}$ was used to define genome-wide significance. Clumping was performed (in PLINK 1.90[51]) to identify independent loci. We formed clumps of all SNPs which were ±500kb of each index SNP with a linkage disequilibrium $r^2 > 0.001$. Only the index SNP within each clump is reported. For multi-ancestry index variants within 500 kb of index SNPs identified in the European-only analysis, we considered these to be independent if the lead multi-ancestry SNP was not in LD ($r^2 < 0.01$) with the lead neighbouring European variant. Multi-ancestry fixed effect meta-analysis was also performed for comparison with the MR-MEGA results.

### Known/Novel assignment

Novel associations are defined as a SNP that had not been reported in a previous GWAS (Supplementary Data 1), or was not correlated ($r^2 < 0.1$ in the relevant ancestry) with a known SNP from this list. In addition, following the assignment of genes to loci (see gene prioritisation) any locus annotated with a gene that has been previously reported were also moved to the 'known' list. Therefore, some loci which are reported in Open Targets[52,53] (but not reported in a published AD GWAS study) have been classed as novel. These loci are marked as such in Table 2.

### Conditional analysis

Conditional analysis was performed to identify any independent secondary associations in the European meta-analysis. Genome-wide

complex trait analysis-conditional and joint analysis (GCTA-COJO[54]) was used to test for independent associations 250 kb either side of the index SNPs using UK Biobank HRC imputed data as the reference. COJO-slct was used to determine which SNPs in the region were conditionally independent (using default $P < 1 \times 10^{-5}$) and therefore represent independent secondary associations. COJO-cond was then used to condition on the top hit in each region to determine the conditional effect estimates.

### Replication

The genome-wide index SNPs identified from the European and mixed-ancestry discovery meta-analyses were taken forward for replication in 23andMe, Inc. Individuals of European ($N = 2,904,664$), Latino ($N = 525,348$) and African ancestry ($N = 174,015$) were analysed separately. Full details are available in the Supplementary Methods.

### LD score regression

Linkage disequilibrium score (LDSC) regression software (version 1.0.1)[55] was used to estimate the SNP-based heritability ($h^2_{SNP}$) for AD. This was performed with the summary statistics of the European discovery meta-analysis. The $h^2_{SNP}$ was estimated on liability scale with a population prevalence of 0.15 and a sample prevalence of 0.070.

Genetic correlation with other traits was assessed using all the traits available on CTG-VL[56] (accessed on 5th November 2021). We considered phenotypes with p-values below the Bonferroni-corrected alpha threshold (i.e., $0.05/1376 = 4 \times 10^{-5}$) to be genetically correlated with AD (a conservative threshold given the likely correlation between many traits tested).

### Bioinformatic analysis

For the following analyses we defined the regions within which the true causal SNP resides to be determined by boundaries containing furthest distanced SNPs with $r^2 >= 0.2$ within ±500kb of the index SNP[18]. We refer to such regions as locus intervals and we used them as input for the analyses described below.

### Enrichment analysis

Enrichment of tissues and cell types and gene sets for AD GWAS loci was investigated using DEPICT[57] and GARFIELD (GWAS analysis of regulatory or functional information enrichment with LD correction)[58] ran with default settings, as well as MAGMA v.1.06[59] (using GTEx ver. 8[23] on the FUMA[60] platform). In addition, we used MendelVar[61] run with default settings to check for enrichment of any ontology terms assigned to Mendelian disease genes within the locus interval regions.

By default, MAGMA only assigns variants within genes. DEPICT maps all genes within a given LD ($r^2 > 0.5$) boundary of the index variant. DEPICT gene set enrichment results for GO terms only were grouped (using the Biological Processes ontology) and displayed using the rrvgo package. The default scatter function was adapted to only plot parent terms[62].

### Prioritisation of candidate genes

To prioritise candidate genes at each of the loci identified in the European GWAS, we investigated all genes within ±500 kb of each index SNP (selected to capture an estimated 98% of causal genes)[63]. The approach used has been previously described by Sobczyk et al.[18]. For each gene we collated evidence from a range of approaches (as described below) to link SNP to gene, resulting in 14 annotation categories (represented as columns in Supplementary Fig. 7). We summarised these annotations for each gene into a score in order to prioritise genes at each locus. We present the top prioritised gene in the main tables, but strength of evidence varies and so we encourage readers to use our full evaluation (of all the evidence presented in Supplementary Data 11 for all genes at each locus) for loci of interest.

We tested for colocalisation with molecular QTLs, where full summary statistics were available, using coloc[64] method (with betas as input). We used the eQTL Catalogue[65] and Open GWAS[66] to download a range of eQTL datasets from all skin, whole blood and immune cell types as well as additional tissue types which showed enrichment for our GWAS loci, such as spleen and oesophagus mucosa[18]. A complete list of eQTL datasets[20–23,26–31,33,67–71] is displayed in Supplementary Data 13. pQTL summary statistics for plasma proteins[72] were downloaded from Open GWAS. An annotation was included in our gene prioritisation pipeline if there was a posterior probability >95% that the associations from the AD GWAS and the relevant QTL analysis shared the same causal variant.

Additional colocalisation methods were also applied. TWAS (Transcriptome-Wide association Study)-based S-MultiXcan[73] and SMR (Summary-based Mendelian Randomization)[74] were run on datasets available via the CTG-VL platform (including GTEx tissue types and 2 whole blood pQTL[72,75] datasets available for the SMR pipeline). For S-MultiXcan and SMR, we report only results with p-values below the alpha threshold established with Bonferroni correction, as well as no evidence of heterogeneity (HEIDI $P$-value > 0.05) in SMR analysis.

Genes were also annotated if they were included in any of the globally enriched ontology/pathway terms from the MendelVar analysis described above or if they were identified in direct look-ups of keywords: "skin", "kera", "derma" in their OMIM[76] descriptions, or Human Phenotype Ontology[77]/Disease Ontology[78] terms.

We also used machine learning candidate gene prioritisation pipelines – DEPICT[57], PoPs[79], POSTGAP[80] and Open Targets Genetics[53] Variant 2 Gene mapping tool as well as gene-based MAGMA[59] test. We added annotations to genes reported in the top 3 (by each of the pipelines).

We mined the literature for a list of differential expression studies and found 9 RNA-Seq/microarray plus 4 proteomic analyses involving comparisons of AD lesional[25,32,81–84] or AD nonlesional[24,25,32,82,85–87] skin vs healthy controls. Studies with comparisons of AD lesional acute vs chronic[88], blood proteome in AD vs healthy control[32] and *FLG* knockdown vs control in living skin-equivalent[89] were also included. We annotated each gene (including direction of effect, i.e. upregulated/ downregulated) with FDR < 0.05 in any dataset.

Lastly, we annotated genes where the index SNP resided within the coding region according to VEP (Variant Effect Predictor)[90] analysis.

For each candidate gene, we established a pragmatic approach to combine all available evidence in order to prioritise which the most plausible candidate gene(s). This prioritisation was carried out as follows:

- The number of annotations (each representing one piece of evidence) were summed across all methods and datasets, to derive a 'total evidence score', i.e., if coloc evidence was observed for 5 datasets for a particular gene, this would add 5 to the score for that gene.
- Additionally, to assess if evidence was coming from multiple datasets using the same method, or evidence was coming from diverse approaches, we counted 'evidence types', summing up the methods (as opposed to datasets) with an annotation for each gene tested (up to a maximum of 14), i.e., in the same example of coloc evidence observed in 5 datasets, this would add 1 to this measure for this gene. Evidence types are represented by the columns in Supplementary Fig. 7.
- In order to prioritise genes with the most evidence, whilst ensuring there was some evidence of triangulation across methods, at each locus we prioritised the gene with the highest 'total evidence score' with a minimum 'evidence type' of 3. 'Evidence type' was also used to break ties.

### Network analysis

Network analysis of the prioritised genes was carried out using standard settings (minimum interaction score 0.4) in STRING v11.5[91].

### Reporting summary

Further information on research design is available in the Nature Portfolio Reporting Summary linked to this article.

## Data availability

Summary statistics of the GWAS meta-analyses generated in this study have been deposited in the GWAS Catalog under study accession IDs GCST90244787 and GCST90244788. The variant-level data for the 23andMe replication dataset are fully disclosed in the main tables and supplementary tables. Individual-level data are protected and are not available due to data privacy laws, and in accordance with the IRB-approved protocol under which the study was conducted.

## Code availability

Code for the bioinformatic analysis is available here: https://github.com/marynias/eczema_gwas_fu/tree/bc4/new_gwas.

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

## Acknowledgements

For this work, A.B.-A., S.J.B. and L.P. were funded by the Innovative Medicines Initiative 2 Joint Undertaking (JU) under grant agreement No. 821511 (BIOMAP). The J.U. receives support from the European Union's Horizon 2020 research and innovation programme and EFPIA. This publication reflects only the author's view and the J.U. is not responsible for any use that may be made of the information it contains. A.B.A., M.K.S., J.L.M., and L.P. and work in a research unit funded by the UK Medical Research Council (MC_UU_00011/1 and MC_UU_00011/4). LP received funding from the British Skin Foundation (8010 Innovative Project) and the Academy of Medical Sciences Springboard Award, which is supported by the Wellcome Trust, The Government Department for Business, Energy and Industrial Strategy, the Global Challenges Research Fund and the British Heart Foundation [SBF003\1094]. S.J.B. holds a Wellcome Trust Senior Research Fellowship in Clinical Science [220875/Z/20/Z]. S.H. is supported by a Vera Davie Study and Research Sabbatical Bursary, NRF Thuthuka Grant (117721), NRF Competitive Support for Unrated Researcher (138072), MRC South Africa under a Self-initiated grant. M.S. has received funding from the European Research Council (ERC) under the European Union's Horizon 2020 research and innovation programme (Grant Agreement No. 949906). Thanks to Sergi Sayols (developer of rrvgo), who provided additional code to alter the scatter plot produced by rrvgo to only display parent terms, and to Gibran Hemani (University of Bristol) who provided valuable guidance on the comparison of effects between ancestries. This publication is the work of the authors and LP will serve as guarantor for the contents of this paper. This work was carried out using the computational facilities of the Advanced Computing Research Centre, University of Bristol—http://www.bristol.ac.uk/acrc/. Individual cohort acknowledgements are in the Supplementary Methods.

## Author contributions

Designed and co ordinated the study: M.St., L.P. Performed the meta analysis: A.B.-A., A. Ki. Performed the bioinformatic analysis: M.K.So. Performed the STRING analysis: S.J. Br. Performed statistical analysis within cohorts: A.B.-A., A. Ki, R. Mi, K.R., R. Mä, M.N., N.T., B.M.B., L.F.T., P.S.N., C.F., A.E.O., E.H.L., J.V.T.L., J.B.J., I.M., A.A.-S., A.J., H. Ba., E.R., A. Ku., C.M.G., C.H., C.Q., P.T., E.A., J.F., C.A.W., E.T., B.W., S.K., D.M.K., L.K., J.D., H.Z., C.A., V.U., R.K., A. Sz., A.C.S.N.J., A.G., M.I., M. M.-Nu., T.S.A., M.B., C.G., M.P.Y., D.P.S., N.G., Y.A.L., A.D.I., L.K.W., C.M., S.J. Br. Data acquisition/supported analysis/interpretation of data: A.B.-A., A. Ki, A.R., P.S.N., A.J., C.j.X., S.E.H., J.F., E.T., S.K., H.Z., S.H., T. Ho., E.J., H.C., N.R., P.N., O.A.A., S.J., C.A., T.G., V.U., P.K.E.M., E.G.B., J.P.T., T. Ha., L.L.K., T.M.D., A.Ar., G.H., S.L., M.M. Nö., N.H., M.I., T.S.Ah., J.S., B.C., A.M.M.S., A.E., S.Ar., T.E., L.A.M., A.M., C.T., K.A., M.L., K.H., B.J., D.P.S., Y.A.L., N.P.H., S.W., A.D.I., D.J., T.N., L.D., J.M.V., G.H.K., K.M.G., B.F., C.E.P., P.D.S., P.G.Ho., H. Bi., K.B., J.C., A. Si., T.S., S.J. Br., M.St., L.P. Wrote the paper: A.B.-A., A. Ki, M.K.So., S.J. Br., M.S., L.P. Approved final version of paper: A.B.-A., A. Ki, M.K.So., S.S.S., R. Mi, K.R., A.R., R. Mä, M.N., N.T., B.M.B., L.F.T., P.S.N., C.F., A.E.O., E.H.L., J.V.T.L., J.B.J., I.M., A.A.-S., A.J., K.A.F., H. Ba., E.R., A.C.A., A. Ku., P.M.S., X.C., C.M.G., C.H., C.j.X., C.Q., S.E.H., P.T., E.A., J.F., C.A.W., E.T., B.W., S.K., D.M.K., L.K., J.D., H.Z., S.H., T. Ho., E.J., H.C., N.R., P.N., O.A.A., S.J., C.A., T.G., V.U., R.K., P.K.E.M., A. Sz., E.G.B., J.P.T., T. Ha., L.L.K., T.M.D., A.C.S.N.J., A.G., A.Ar., G.H., S.L., M.M. Nö., N.H., M.I., A.V., M.F., V.B., P.Hy., N.B., D.I.B., J.J.H., M.M.-Nu., T.S.Ah., J.S., B.C., A.M.M.S., A.E., M.B., B.R., S.Ar., C.G., T.E., L.A.M., A.M., C.T., K.A., M.L., K.H., B.J., M.P.Y., D.P.S., N.G., A.L., Y.A.L., N.P.H., S.W., M.R.J., E.M., H.H., A.D.I., D.J., T.N., L.D., J.M.V., G.H.K., K.M.G., S.J. Ba., B.F., C.E.P., P.D.S., P.G.Ho., L.K.W., H. Bi., K.B., J.C., A. Si., C.M., T.S., S.Bu., S.T.W., J.W.H., J.L.M., S.J. Br., M.St., L.P.

## Competing interests

K.M.G. has received reimbursement for speaking at conferences sponsored by companies selling nutritional products, and is part of an academic consortium that has received research funding from Abbott Nutrition, Nestec, BenevolentAI Bio Ltd. and Danone. C.G., S.S.S., and 23andMe Research Team are employed by and hold stock or stock options in 23andMe, Inc. The remaining authors declare no competing interests.

## Additional information

Ashley Budu-Aggrey[1,2,124], Anna Kilanowski[3,4,5,124], Maria K. Sobczyk[1,2], 23andMe Research Team, Suyash S. Shringarpure[6], Ruth Mitchell[1,2], Kadri Reis[7], Anu Reigo[7], Estonian Biobank Research Team, Reedik Mägi[7], Mari Nelis[7,8], Nao Tanaka[9,10], Ben M. Brumpton[11,12,13], Laurent F. Thomas[11,14,15,16], Pol Sole-Navais[17], Christopher Flatley[17], Antonio Espuela-Ortiz[18], Esther Herrera-Luis[18], Jesus V. T. Lominchar[19], Jette Bork-Jensen[19], Ingo Marenholz[20,21], Aleix Arnau-Soler[20,21], Ayoung Jeong[22,23], Katherine A. Fawcett[24], Hansjorg Baurecht[25], Elke Rodriguez[26], Alexessander Couto Alves[27], Ashish Kumar[28], Patrick M. Sleiman[29,30,31], Xiao Chang[29], Carolina Medina-Gomez[32,33], Chen Hu[32,34], Cheng-jian Xu[35,36,37,38], Cancan Qi[35,36], Sarah El-Heis[39], Philip Titcombe[39], Elie Antoun[40,41], João Fadista[42,43,44,45], Carol A. Wang[46,47], Elisabeth Thiering[3,4], Baojun Wu[48], Sara Kress[49], Dilini M. Kothalawala[50,51], Latha Kadalayil[50], Jiasong Duan[52], Hongmei Zhang[52], Sabelo Hadebe[53], Thomas Hoffmann[54,55], Eric Jorgenson[56], Hélène Choquet[57], Neil Risch[54,55], Pål Njølstad[58,59], Ole A. Andreassen[60,61], Stefan Johansson[58,62], Catarina Almqvist[63,64], Tong Gong[63], Vilhelmina Ullemar[63], Robert Karlsson[63], Patrik K. E. Magnusson[63], Agnieszka Szwajda[63], Esteban G. Burchard[65,66], Jacob P. Thyssen[67], Torben Hansen[19], Line L. Kårhus[68], Thomas M. Dantoft[68], Alexander C.S.N. Jeanrenaud[20,21], Ahla Ghauri[20,21], Andreas Arnold[69], Georg Homuth[70], Susanne Lau[71], Markus M. Nöthen[72], Norbert Hübner[20,73], Medea Imboden[22,23], Alessia Visconti[74], Mario Falchi[74], Veronique Bataille[74,75], Pirro Hysi[74], Natalia Ballardini[28], Dorret I. Boomsma[76,77], Jouke J. Hottenga[76], Martina Müller-Nurasyid[78,79,80], Tarunveer S. Ahluwalia[81,82,83], Jakob Stokholm[81,84], Bo Chawes[81], Ann-Marie M. Schoos[81,84], Ana Esplugues[85,86], Mariona Bustamante[87,88,89], Benjamin Raby[90], Syed Arshad[91,92], Chris German[6], Tõnu Esko[7], Lili A. Milani[7], Andres Metspalu[7], Chikashi Terao[9,93,94], Katrina Abuabara[95], Mari Løset[11,96], Kristian Hveem[11,97], Bo Jacobsson[17,98], Maria Pino-Yanes[18,99,100], David P. Strachan[101], Niels Grarup[19], Allan Linneberg[68,102], Young-Ae Lee[20,21], Nicole Probst-Hensch[22,23], Stephan Weidinger[103], Marjo-Riitta Jarvelin[104,105,106], Erik Melén[28], Hakon Hakonarson[29,107,108], Alan D. Irvine[109], Deborah Jarvis[110,111], Tamar Nijsten[34], Liesbeth Duijts[112,113], Judith M. Vonk[36,114], Gerard H. Koppelman[35,36], Keith M. Godfrey[115], Sheila J. Barton[39], Bjarke Feenstra[43], Craig E. Pennell[46,47], Peter D. Sly[116,117], Patrick G. Holt[118], L. Keoki Williams[48], Hans Bisgaard[81,126], Klaus Bønnelykke[81], John Curtin[119], Angela Simpson[119], Clare Murray[119], Tamara Schikowski[120], Supinda Bunyavanich[121], Scott T. Weiss[90], John W. Holloway[50,91], Josine L. Min[1,2], Sara J. Brown[122], Marie Standl[3,123] & Lavinia Paternoster[1,2,125]✉

[1]Medical Research Council Integrative Epidemiology Unit, Bristol Medical School, University of Bristol, Bristol, England. [2]Population Health Sciences, Bristol Medical School, University of Bristol, Bristol, England. [3]Institute of Epidemiology, Helmholtz Zentrum München - German Research Center for Environmental Health, Neuherberg, Germany. [4]Division of Metabolic and Nutritional Medicine, Dr. von Hauner Children's Hospital, University of Munich Medical Center, Munich, Germany. [5]Pettenkofer School of Public Health, Ludwig-Maximilians University Munich, Munich, Germany. [6]23andMe, Inc., Sunnyvale, CA, USA. [7]Estonian Genome Centre, Institute of Genomics, University of Tartu, Tartu, Estonia. [8]Core Facility of Genomics, University of Tartu, Tartu, Estonia. [9]Laboratory for Statistical and Translational Genetics, RIKEN Center for Integrative Medical Sciences, Yokohama, Japan. [10]Department of Rheumatology, Graduate School of Medical and Dental Sciences, Tokyo Medical and Dental University (TMDU), Tokyo, Japan. [11]K.G. Jebsen Center for Genetic Epidemiology, Department of Public Health and Nursing, NTNU, Norwegian University of Science and Technology, Trondheim 7030, Norway. [12]HUNT Research Centre, Department of Public Health and Nursing, NTNU, Norwegian University of Science and Technology, Levanger 7600, Norway. [13]Clinic of Medicine, St. Olavs Hospital, Trondheim University Hospital, Trondheim 7030, Norway. [14]Department of Clinical and Molecular Medicine, NTNU Norwegian University of Science and Technology, Trondheim, Norway. [15]BioCore - Bioinformatics Core Facility, NTNU, Norwegian University of Science and Technology, Trondheim, Norway. [16]Clinic of Laboratory Medicine, St. Olavs Hospital, Trondheim University Hospital, Trondheim, Norway. [17]Department of Obstetrics and Gynecology, Institute of Clinical Sciences, Sahlgrenska Academy, University of Gothenburg, Gothenburg, Sweden. [18]Genomics and Health Group, Department of Biochemistry, Microbiology, Cell Biology and Genetics, Universidad de La Laguna, La Laguna, Tenerife, Spain. [19]Novo Nordisk Foundation Center for Basic Metabolic

Research, Faculty of Health and Medical Sciences, University of Copenhagen, København, Denmark. [20]Max-Delbrück-Center for Molecular Medicine, Berlin, Germany. [21]Clinic for Pediatric Allergy, Experimental and Clinical Research Center, Charité-Universitätsmedizin Berlin, Berlin, Germany. [22]Swiss Tropical and Public Health Institute, CH-4123 Basel, Switzerland. [23]University of Basel, CH-4001 Basel, Switzerland. [24]Department of Health Sciences, University of Leicester, Leicester LE1 7RH, UK. [25]Department of Epidemiology and Preventive Medicine, University of Regensburg, Regensburg, Germany. [26]Department of Dermatology and Allergy, University Hospital Schleswig-Holstein, Kiel, Germany. [27]School of Biosciences and Medicine, University of Surrey, Guildford, UK. [28]Department of Clinical Science and Education Södersjukhuset, Karolinska Institutet, Solna, Sweden. [29]Center for Applied Genomics, Children's Hospital of Philadelphia, Philadelphia, PA 19104, USA. [30]Department of Genetics, Perelman School of Medicine, University of Pennsylvania, Philadelphia, USA. [31]Rhythm Pharmaceuticals, 222 Berkley Street, Boston 02116, USA. [32]The Generation R Study Group, Erasmus MC, University Medical Center Rotterdam, Rotterdam, The Netherlands. [33]Department of Internal Medicine, Erasmus MC, University Medical Center Rotterdam, Rotterdam, The Netherlands. [34]Department of Dermatology, Erasmus MC, University Medical Center Rotterdam, Rotterdam, The Netherlands. [35]University of Groningen, University Medical Center Groningen, Department of Pediatric Pulmonology and Pediatric Allergy, Beatrix Children's Hospital, Groningen, The Netherlands. [36]University of Groningen, University Medical Center Groningen, GRIAC Research Institute, Groningen, The Netherlands. [37]Centre for Individualized Infection Medicine, CiiM, a joint venture between Hannover Medical School and the Helmholtz Centre for Infection Research, Hannover, Germany. [38]TWINCORE, Centre for Experimental and Clinical Infection Research, a joint venture between the Hannover Medical School and the Helmholtz Centre for Infection Research, Hannover, Germany. [39]MRC Lifecourse Epidemiology Centre, University of Southampton, Southampton, UK. [40]Faculty of Medicine, University of Southampton, Southampton, UK. [41]Institute of Developmental Sciences, University of Southampton, Southampton, UK. [42]Department of Bioinformatics & Data Mining, Måløv, Denmark. [43]Department of Epidemiology Research, Statens Serum Institut, Copenhagen, Denmark. [44]Department of Clinical Sciences, Lund University Diabetes Centre, Malmö, Sweden. [45]Institute for Molecular Medicine Finland (FIMM), University of Helsinki, Helsinki, Finland. [46]School of Medicine and Public Health, University of Newcastle, Newcastle, NSW, Australia. [47]Hunter Medical Research Institute, Newcastle, NSW, Australia. [48]Center for Individualized and Genomic Medicine Research (CIGMA), Department of Medicine, Henry Ford Health, Detroit, MI 48104, USA. [49]Environmental Epidemiology of Lung, Brain and Skin Aging, IUF – Leibniz Research Institute for Environmental Medicine, Düsseldorf, Germany. [50]Human Development and Health, Faculty of Medicine, University of Southampton, Southampton, UK. [51]NIHR Southampton Biomedical Research Centre, University Hospital Southampton, Southampton, UK. [52]Division of Epidemiology, Biostatistics, and Environmental Health, School of Public Health, University of Memphis, Memphis, TN, USA. [53]Division of Immunology, Department of Pathology, Faculty of Health Sciences, University of Cape Town, Cape Town, South Africa. [54]Institute for Human Genetics, UCSF, San Francisco, CA 94143, USA. [55]Department of Epidemiology and Biostatistics, UCSF, San Francisco, CA 94158, USA. [56]Regeneron Genetics Center, Tarrytown, NY, USA. [57]Division of Research, Kaiser Permanente Northern California, Oakland, CA, USA. [58]Center for Diabetes Research, Department of Clinical Science, University of Bergen, NO-5020 Bergen, Norway. [59]Children and Youth Clinic, Haukeland University Hospital, NO-5021 Bergen, Norway. [60]NORMENT Centre, Institute of Clinical Medicine, University of Oslo, 0450 Oslo, Norway. [61]Division of Mental Health and Addiction, Oslo University Hospital, 0450 Oslo, Norway. [62]Department of Medical Genetics, Haukeland University Hospital, NO-5021 Bergen, Norway. [63]Department of Medical Epidemiology and Biostatistics, Karolinska Institutet, Stockholm, Sweden. [64]Pediatric Lung and Allergy Unit, Astrid Lindgren Children's Hospital, Karolinska University Hospital, Stockholm, Sweden. [65]Department of Medicine, University of California San Francisco, San Francisco, CA, USA. [66]Department of Bioengineering and Therapeutic Sciences, University of California San Francisco, San Francisco, CA, USA. [67]Department of Dermatology, Bispebjerg Hospital, University of Copenhagen, Copenhagen, Denmark. [68]Center for Clinical Research and Prevention, Bispebjerg and Frederiksberg Hospital, Frederiksberg, Denmark. [69]Clinic and Polyclinic of Dermatology, University Medicine Greifswald, Greifswald, Germany. [70]Department of Functional Genomics, Interfaculty Institute for Genetics and Functional Genomics, University Medicine Greifswald, Greifswald, Germany. [71]Department of Pediatric Respiratory Medicine, Immunology, and Critical Care Medicine, Charité-Universitätsmedizin Berlin, Berlin, Germany. [72]Institute of Human Genetics, University of Bonn, School of Medicine & University Hospital Bonn, Bonn, Germany. [73]Charite-Universitätsmedizin Berlin, Berlin, Germany. [74]Department of Twin Research & Genetics Epidemiology, Kings College London, London, UK. [75]Dermatology Department, West Herts NHS Trust, Watford, UK. [76]Dept Biological Psychology, Netherlands Twin Register, VU University, Amsterdam, the Netherlands. [77]Institute for Health and Care Research (EMGO), VU University, Amsterdam, the Netherlands. [78]Institute of Genetic Epidemiology, Helmholtz Zentrum München - German Research Center for Environmental Health, Neuherberg, Germany. [79]IBE, Faculty of Medicine, LMU Munich, Munich, Germany. [80]Institute of Medical Biostatistics, Epidemiology and Informatics (IMBEI), University Medical Center, Johannes Gutenberg University, Mainz, Germany. [81]COPSAC, Copenhagen Prospective Studies on Asthma in Childhood, Herlev and Gentofte Hospital, University of Copenhagen, Copenhagen, Denmark. [82]Steno Diabetes Center Copenhagen, Herlev, Denmark. [83]Department of Biology, University of Copenhagen, Copenhagen, Denmark. [84]Department of Pediatrics, Slagelse Hospital, Slagelse, Denmark. [85]Nursing School, University of Valencia, FISABIO-University Jaume I-University of Valencia, Valencia, Spain. [86]Joint Research Unit of Epidemiology and Environmental Health, CIBERESP, Valencia, Spain. [87]ISGlobal, Institute for Global Health, Barcelona, Spain. [88]Universitat Pompeu Fabra (UPF), Barcelona, Spain. [89]CIBER Epidemiología y Salud Pública, Madrid, Spain. [90]Channing Division of Network Medicine, Brigham & Women's Hospital and Harvard Medical School, Boston, MA, USA. [91]Clinical and Experimental Sciences, Faculty of Medicine, University of Southampton, Southampton, UK. [92]David Hide Asthma and Allergy Research Centre, Isle of Wight, UK. [93]Clinical Research Center, Shizuoka General Hospital, Shizuoka, Japan. [94]Department of Applied Genetics, School of Pharmaceutical Sciences, University of Shizuoka, Shizuoka, Japan. [95]Department of Dermatology, University of California San Francisco, San Francisco, CA, USA. [96]Department of Dermatology, Clinic of Orthopaedy, Rheumatology and Dermatology, St. Olavs Hospital, Trondheim University Hospital, Trondheim, Norway. [97]HUNT Research Centre, Department of Public Health and General Practice, Norwegian University of Science and Technology, Levanger, Norway. [98]Department of Genetics and Bioinformatics, Norwegian Institute of Public Health, Oslo, Norway. [99]CIBER de Enfermedades Respiratorias, Instituto de Salud Carlos III, Madrid, Spain. [100]Instituto de Tecnologías Biomédicas (ITB), Universidad de La Laguna, San Cristóbal de La Laguna, Santa Cruz de Tenerife, Spain. [101]Population Health Research Institute, St George's, University of London, Cranmer Terrace, London SW17 0RE, UK. [102]Department of Clinical Medicine, Faculty of Health and Medical Sciences, University of Copenhagen, Copenhagen, Denmark. [103]Department of Dermatology, Allergology and Venereology, University Hospital Schleswig-Holstein, Kiel, Germany. [104]Department of Epidemiology and Biostatistics, MRC-PHE Centre for Environment & Health, School of Public Health,Imperial College London, London, UK. [105]Center for Life Course Health Research, Faculty of Medicine, University of Oulu, Oulu, Finland. [106]Biocenter Oulu, University of Oulu, Oulu, Finland. [107]Department of Pediatrics, Divisions of Human Genetics and Pulmonary Medicine, Perelman School of Medicine, University of Pennsylvania, Philadelphia, PA 19104, USA. [108]Faculty of Medicine, University of Iceland, 101 Reykjavík, Iceland. [109]Department of Clinical Medicine, Trinity College, Dublin, Ireland. [110]Respiratory Epidemiology, Occupational Medicine and Public Health, National Heart and Lung Institute, Imperial College London, London, United Kingdom. [111]Medical Research Council and Public Health England Centre for Environment and Health, London, United Kingdom. [112]Department of Pediatrics, division of Respiratory Medicine and Allergology, Erasmus MC, University Medical Center Rotterdam, Rotterdam, The Netherlands. [113]Department of Pediatrics, division of Neonatology, Erasmus MC, University Medical Center Rotterdam, Rotterdam, The Netherlands. [114]University of Groningen, University Medical Center Groningen, Department of Epidemiology, Groningen, The Netherlands. [115]MRC Lifecourse Epidemiology Centre and

NIHR Southampton Biomedical Research Centre, University of Southampton and University Hospital Southampton NHS Foundation Trust, Southampton, UK. [116]Children's Health and Environment Program, Child Health Research Centre, The University of Queensland, South Brisbane 4101 Queensland, Australia. [117]Australian Infectious Diseases Research Centre, The University of Queensland, St Lucia 4072 QLD, Australia. [118]Telethon Kids Institute, University of Western Australia, Perth, WA, Australia. [119]Division of Immunology, Immunity to Infection and Respiratory Medicine, School of Biological Sciences, The University of Manchester, Manchester Academic Health Science Centre, and Manchester University NHS Foundation Trust, Manchester, England. [120]Environmental Epidemiology of Lung, Brain and Skin Aging, Leibniz Research Institute for Environmental Medicine, Düsseldorf, Germany. [121]Division of Allergy and Immunology, Department of Pediatrics, and Department of Genetics and Genomic Sciences, Icahn School of Medicine at Mount Sinai, New York, NY, USA. [122]Centre for Genomics and Experimental Medicine, Institute for Genetics and Cancer, University of Edinburgh, Crewe Road, Edinburgh UK EH4 2XU, Scotland. [123]German Center for Lung Research (DZL), Munich, Germany. [124]These authors contributed equally: Ashley Budu-Aggrey, Anna Kilanowski. [125]These authors jointly supervised this work: Marie Standl, Lavinia Paternoster. [126]Deceased: Hans Bisgaard.
✉e-mail: l.paternoster@bristol.ac.uk

## 23andMe Research Team

**Suyash S. Shringarpure** ® [6] **& Chris German**[6]

## Estonian Biobank Research Team

**Reedik Mägi** ® [7]**, Mari Nelis**[7,8]**, Tõnu Esko**[7]**, Lili A. Milani** ® [7] **& Andres Metspalu** ® [7]

