## [Peer Review File · Nature Communications]

European and multi-ancestry genome-wide association meta-analysis of atopic dermatitis highlights importance of systemic immune regulationREVIEWER COMMENTS

Reviewer #1 (Remarks to the Author):

This study represents a major effort by several groups and leaders in the field of atopic dermatitis with the largest GWAS to-date in a multi-ethnic cohort. The authors report confirmation of most of the effects previously reported previous with the addition of some novel effects, particularly in non-European ancestries. They report that most effects were common across ancestry and that those unique to a given people-group were not due to differences in frequency.

In general the the methods of study are clearly written and conveys results that will be informative for future fine-mapping and functional studies. That said, the results seem somewhat incremental and how to translate the findings into "next-steps" is lacking. For example, while the authors discuss the need for future studies to classify these effects by clinical manifestation, age of onset, etc, to thus define interventional strategies, but it is not clear why this was not done in this study. Granted, the replication cohort likely had only the self-report diagnosis, but presumably these data would be available for the combined discovery cohort. As another example, the authors discuss the allele frequency differences for variants found in one population and not in another, but never mention the role rare variants not captured by the genotyping platform (or imputable) might play in disease, nor that rare variants might explain why the non-EU populations had far fewer significant variants.

A few other points:

A discussion of the previously identified effects that were not replicated would be welcome to perhaps address points made above about genetic effects specific to sub-phenotypes or rare variants. The summary of results, which is primarily in terms of SNPs would benefit from inclusion of genes. For example, on page 13 the authors state "...differences in associations between ancestries do not seem to be driven by different allele frequencies between populations..." and it is stated that the criteria for defining "novel" was based on $r^2 < 0.1$. However, it wasn't discussed if variants, not in LD, found in the same gene were found in different populations and thus potentially the same genomic consequence.

Similarly, Figure 4 of the predicted protein interaction network would be much better summarized as pathways or at least smaller clusters of genes. As it is, Figure 4 is uninterpretable.

Figure 2 B is also not particularly informative over and above what is already presented in Panel A

Figure 3 could also benefit from more explanation or clarification. For example, when a gene is listed more than once, are the SNPs include the only significant SNPs found? It is the assumption that is not the case, but if not then why?

Overall, the manuscript would greatly benefit from a more clear translation of the results and a reduction in the details presented. As stated, the results will be informative for additional studies to come, but, as presented, seem incremental.

Reviewer #2 (Remarks to the Author):

Summary:

Aggrey et al conducted AD-GWAS meta-analysis in European ancestry and then extend the analysis to multi-ancestry scale. They identified 29 novel loci in European-only analysis and 6 novel loci in multi-ancestry analysis. They conducted standard GWAS downstream analyses (e.g., epigenomic marks, eQTL colocalization, pathway analysis) and found plausible results. Finally, they prioritized candidate genes by integrating multiomic data. Although I believe their findings will be valuable resources for future genetic studies, there are multiple technical concerns that should be addressed.

Major comments:

1, rs77869365 has opposite allelic direction between Europeans and Japanese. In Sup-Table 4, OR in EUR is 1.06 (1.03-1.08) and OR in JPT is 0.69 (0.59-0.81). If it reflects true biology, it should be a very important finding in this article since one of the scopes of this study is to find ancestry-specific

signals. If it does not, the authors should provide the potential reason why this happened.

2, Lines 279-281: "Four SNPs which did not replicate in any of the samples (rs9864845, rs34665982, rs45602133, rs4312054) appeared to have been driven by association in the Japanese RIKEN study only (Supplementary Table 4, Supplementary Figure 2)" is concerning. I have several comments on this finding.

2-1: For these SNPs, the authors need to provide sufficient investigation to exclude they are not artifact, and truly reflect ancestry specific signals (I noticed they provided some comments about these SNPs in the discussion section). If the loci have different LD structures across ancestries, the lead SNPs in JPT studies may not be the best SNP to test the absence of signals in other ancestries. The authors could provide locuszoom (or similar) plots for these loci for each ancestry GWAS. The authors have to show that these "loci" (not only at these lead "SNPs") lack signals in non-JPT ancestries.

2-2: The authors used MR-MEGA which accounts for the heterogeneity in the effect size estimates across ancestries. However, the substantially heterogeneous associations at rs77869365 and these four SNPs indicates MR-MEGA approach maybe too permissive about the heterogeneous signals. The authors should report statistics using an inverse variance-weighted fixed-effect meta-analysis and provide sufficient discussion on the discrepancies (if any) between MR-MEGA and fixed-effect meta-analysis approach.

3, Lines 282-283: "A further 4 SNPs did not replicate, and on closer examination (Supplementary Figure 2, and MAF in cases <1%), their association in the discovery analysis appeared to be driven by a false positive outlying result in a single European cohort." This is also very concerning. This indicates that MR-MEGA approach may also be too permissive about the heterogeneous signals even within a single ancestry. How do the fixed-effect statistics behave at these SNPs? I expect that fixed-effect meta-analysis approach provides more conservative statistics (meaning less false positive signals) compared with MR-MEGA.

4, The authors used 'total evidence score' to prioritize the candidate gene. I understand their motivation very well. However, any scoring system needs calibration. Their strategy is too arbitrary to be used in a scientific journal. They need to provide analyses such as those reported in the OpenTarget manuscript (PMID: 34711957). If they can't provide sufficient analyses on this topic, they need to rely on previously-reported methods whose validity was confirmed in peer-reviewed journals.

Minor comments:

1, The authors should report the number of cases in addition to the total sample size in the main text where they explain GWAS study design (e.g., L260 on page 8). Many recent GWASs have the control samples disproportionately more than case samples. If we only report total sample size, we might overemphasize the study scale.

2, Criteria of novel loci: The authors wrote "Novel loci are defined as a SNP that had not been reported in a previous GWAS (Supplementary Table1), or was not correlated ($r^2 < 0.1$) with a known SNP from this list. Which population did they use to calculate r^2 ? Do the authors calculated r^2 in the ancestry in which previous GWAS conducted?"

3, Significant threshold:

3-1. At L264 (page 8), they set significant threshold at $6e-04$, which seems to be 0.05 divided by 81 (the number of target loci in this analysis). This makes sense but need to be explicitly explain in the main text (e.g., " $P < 0.05/81$ tested loci ($=6e-04$)"). Similar modification should be applied to other related parts (e.g., L275 at page 9).

3-2. How did the authors set the significant threshold in the conditional analyses to $1e-05$?

4, Lambda is not an excellent metric to indicate the bias in GWAS results since this can be inflated by polygenic effects (biological effect), not necessarily by the bias. The authors should report the intercept of LDSC.

5, In the Sup-Table 3, the authors should report the r^2 (in European reference panel) between the "Primary SNP" and the "Secondary SNP" to explain the variants are independent.

6, Typo in Table3. At the row of rs7773987 the authors put "***" and the legend says it is about "rs77869365", a different SNP.

REVIEWER COMMENTS

Reviewer #1 (Remarks to the Author):

This study represents a major effort by several groups and leaders in the field of atopic dermatitis with the largest GWAS to-date in a multi-ethnic cohort. The authors report confirmation of most of the effects previously reported previous with the addition of some novel effects, particularly in non-European ancestries. They report that most effects were common across ancestry and that those unique to a given people-group were not due to differences in frequency.

In general the methods of study are clearly written and conveys results that will be informative for future fine-mapping and functional studies.

- That said, the results seem somewhat incremental and how to translate the findings into "next-steps" is lacking. For example, while the authors discuss the need for future studies to classify these effects by clinical manifestation, age of onset, etc, to thus define interventional strategies, but it is not clear why this was not done in this study. Granted, the replication cohort likely had only the self-report diagnosis, but presumably these data would be available for the combined discovery cohort.

We thank the reviewer for their interest in the future studies we have proposed in the paper, however these require extensive efforts and are beyond the scope of this study. However, such work is currently on-going under a separate grant. More broadly, we do not agree that the translation of findings is lacking from this current manuscript. Whilst the association of the identified SNPs with subtypes of disease referred to will be of interest to explore, the follow-up analysis we have done to prioritise genes (and hence potential drug targets) at each locus is arguably the more important translation aspect of this work.

- As another example, the authors discuss the allele frequency differences for variants found in one population and not in another, but never mention the role rare variants not captured by the genotyping platform (or imputable) might play in disease, nor that rare variants might explain why the non-EU populations had far fewer significant variants.

As the reviewer suggests rare variants not captured by the imputation is one possible explanation for the differences seen between populations at a locus. We now explicitly discuss this option on line 484 in our extended discussion of the differences between populations. It is possible that the causal variant may not be captured by the genotyping chip/imputation panel used, or may have been filtered out due to having a low MAF (<0.01). However, of note, in a recent European rare variant study for eczema, there was little difference in the estimated heritability compared to what we have reported, suggesting rare variants are not a big source of missing heritability for Europeans at least. In contrast for non-European populations, the role of rare variants have been far less studied. Such studies will be important in the future of the field.

- A few other points:
A discussion of the previously identified effects that were not replicated would be welcome to perhaps address points made above about genetic effects specific to sub-phenotypes or rare variants.

The only previously identified variants that we were not able to replicate are those that were rare (MAF < 0.01) or were indels which were excluded from our analysis. This was mentioned in the footnote of supplementary table 1 but we now include it in the main text for completeness:

“Review of previous work in this field (Supplementary Table 1) shows that a total of 198 unique variants (across a much smaller number of loci) have been reported to be associated with AD. We found evidence for all but 7 variants of these being nominally associated in the current GWAS (81% in the European and 96% in the multi-ancestry analysis). Variants we did not find to be associated were either rare variants (MAF < 0.01), or insertion/deletion mutations, which were not included in our analysis.”

- The summary of results, which is primarily in terms of SNPs would benefit from inclusion of genes.

Where SNPs are mentioned in the results section we now also report the nearest genes

- For example, on page 13 the authors state "...differences in associations between ancestries do not seem to be driven by different allele frequencies between populations..." and it is stated that the criteria for defining "novel" was based on $r^2 < 0.1$. However, it wasn't discussed if variants, not in LD, found in the same gene were found in different populations and thus potentially the same genomic consequence.

We now refer to 5 examples whereby the index variants do not replicate between populations but there are alternative signals seen within 200-300kb. As we have been unable to conduct comprehensive gene prioritisation for non-European signals (due to lack of necessary molecular data in certain ancestries) it is unclear if these implicate the same genes, but it seems likely.

The following is now added to the text:

“We further examined locus zoom plots and LD matrices (Supplementary Figure 3) of the loci with inconsistent effects between individuals of European and Japanese ancestry (the only two populations for which we had well-powered full summary statistics). We found that at three of the four “Japanese-specific” loci, there were in fact European genome-wide significant signals within 200kb of the Japanese index SNP. Similarly at two of the three “European-specific” loci there was some evidence for an association ($p < 1 \times 10^{-4}$) in the Japanese cohort within 300kb. The LD matrices also showed some differences between populations, together suggesting that some of the differences observed between individuals of different ancestry are likely to represent loci that are shared between populations, but with differences in the underlying causal variants and/or LD structure.”

- Similarly, Figure 4 of the predicted protein interaction network would be much better summarized as pathways or at least smaller clusters of genes. As it is, Figure 4 is uninterpretable.

Thank you for your interest in the network analysis. We agree that Figure 4 does not fully illustrate all aspects of the data but we chose to highlight one of the most significant clusters of predicted protein-protein interactions using the red colour-coding of 28 proteins within the GO term ‘Regulation of immune system process’ (GO:0002682). Additional pathways and smaller clusters of

genes are also listed in full in Supplementary Table 12, allowing detailed review of each predicted network, pathway and cluster. We now include clearer direction to this Supplementary Table from the figure legend.

- Figure 2 B is also not particularly informative over and above what is already presented in Panel A

We agree that both analyses show strong evidence of enrichment in blood, however the two figures do show consistency of results derived from different data sources and analysis methods and so we feel it is valuable to display both. We have however edited the main text and the figure legend to make it clear that these two figures, while consistent in their conclusions, represent separate analyses.

- Figure 3 could also benefit from more explanation or clarification. For example, when a gene is listed more than once, are the SNPs include the only significant SNPs found? It is the assumption that is not the case, but if not then why?

Thank you for pointing out that further clarification is needed here. The SNPs shown in figure 3 represent the **significant independent loci** found in the study. There are situations where two independent variants do implicate the same gene. These are not the only significant SNPs found, but rather the only *independent* significant SNPs found. To improve clarity the term “independent” has been included in the Figure legend, main text and headers of Tables 1 and 2. In addition we now explicitly acknowledge in the Figure legend that in some cases multiple independent GWAS signals implicate the same gene.

Overall, the manuscript would greatly benefit from a more clear translation of the results and a reduction in the details presented. As stated, the results will be informative for additional studies to come, but, as presented, seem incremental.

Thank you for this comment. Whilst not the first GWAS of atopic dermatitis, we believe this current work represents much more than an incremental increase in knowledge. The GWAS presented is by far the largest ever conducted for eczema. Taking the number of variants reported in a single GWAS study from 31 to 96, representing a huge increase in AD susceptibility loci. We also fully characterise these loci with molecular data and other useful annotations to provide a comprehensive prioritisation of genes at these loci, something which has not to date featured in the AD GWAS papers to this extent. In addition to these novel insights, which provide opportunities for novel drug development, this comprehensive GWAS will be made publicly available to enable to inclusion of AD in a vast range of studies that require powerful GWAS summary statistics, such as Mendelian Randomization to investigate causal effects and investigations of genomic architecture with e.g. LD score regression. We have now expanded the final paragraph of the paper to highlight the value of this important addition to the field. The final paragraph now reads:

“We have presented the largest GWAS of AD to date, identifying 96 robustly associated loci, 35 with some evidence of population-specific effects. This represents a significant increase in knowledge of AD genetics compared to previous efforts, taking the number of GWAS hits identified in a single study from

31 to 96 and making available the well-powered summary statistics to enable many future important studies (e.g. Mendelian Randomization to investigate causal relationships). To aid translation of our results, we have undertaken comprehensive post-GWAS analyses to prioritise potentially causal genes at each locus, implicating many immune system genes and pathways and identifying potential novel drug targets.”

Reviewer #2 (Remarks to the Author):

Summary:

Aggrey et al conducted AD-GWAS meta-analysis in European ancestry and then extend the analysis to multi-ancestry scale. They identified 29 novel loci in European-only analysis and 6 novel loci in multi-ancestry analysis. They conducted standard GWAS downstream analyses (e.g., epigenomic marks, eQTL colocalization, pathway analysis) and found plausible results. Finally, they prioritized candidate genes by integrating multiomic data. Although I believe their findings will be valuable resources for future genetic studies, there are multiple technical concerns that should be addressed.

Major comments:

- 1, rs77869365 has opposite allelic direction between Europeans and Japanese. In Sup-Table 4, OR in EUR is 1.06 (1.03-1.08) and OR in JPT is 0.69 (0.59-0.81). If it reflects true biology, it should be a very important finding in this article since one of the scopes of this study is to find ancestry-specific signals. If it does not, the authors should provide the potential reason why this happened.

We agree that this locus is interesting. The interpretation of the different effect directions is unclear. We think it is unlikely that the same underlying genetic change causes an increased risk in one population and a decreased risk in another, therefore, we think the more likely explanation is that this index SNP is not the causal variant and the LD differences between the populations induces the apparent opposite result in an index SNP. We now include a more thorough investigation of all ancestry-inconsistent results (including this SNP), showing the locus zoom plot and LD matrices for these loci. Whilst not conclusive we believe these indicate the LD differences are a likely factor for at least some of these loci (including rs77869365). Such loci will be important to follow-up, especially with analyses to fine-map and link to molecular data in non-European and multi-ancestry settings. We believe this is an important future direction for the field.

We have added this additional text:

“These results all showed consistent direction of effect, except for rs77869365, where the G allele was associated with increased risk of AD in Europeans (OR=1.06, $p=3 \times 10^{-6}$) and decreased risk (OR=0.69, $p=5 \times 10^{-6}$) in the Japanese cohort (Supplementary Table 4).”

“We further examined locus zoom plots and LD matrices (Supplementary Figure 3) of the loci with inconsistent effects between individuals of European and Japanese ancestry (the only two populations for which we had well-powered full summary statistics). We found that at three of the four “Japanese-specific” loci, there were in fact European genome-wide significant signals within

200kb of the Japanese index SNP. Similarly at two of the three “European-specific” loci there was some evidence for an association ($p < 1 \times 10^{-4}$) in the Japanese cohort within 300kb. The LD matrices also showed some differences between populations, together suggesting that some of the differences observed between individuals of different ancestry are likely to represent loci that are shared between populations, but with differences in the underlying causal variants and/or LD structure.”

- 2, Lines 279-281: “Four SNPs which did not replicate in any of the samples (rs9864845, rs34665982, rs45602133, rs4312054) appeared to have been driven by association in the Japanese RIKEN study only (Supplementary Table 4, Supplementary Figure 2)” is concerning. I have several comments on this finding.
- 2-1: For these SNPs, the authors need to provide sufficient investigation to exclude they are not artifact, and truly reflect ancestry specific signals (I noticed they provided some comments about these SNPs in the discussion section). If the loci have different LD structures across ancestries, the lead SNPs in JPT studies may not be the best SNP to test the absence of signals in other ancestries. The authors could provide locuszoom (or similar) plots for these loci for each ancestry GWAS. The authors have to show that these “loci” (not only at these lead “SNPs”) lack signals in non-JPT ancestries.

Thank you for this comment. We agree that the original conclusions of the results at these loci were weak and we have made several changes to improve this. Whilst we show evidence that these index SNPs are not associated in Europeans, Latin Americans and African Americans, we agree that it is possible that these loci are still important in those populations (but represented by alternative index SNPs). As, included in the response to comment 1, we now include locus zoom plots (supplementary Figure 3) of these regions, which show that for three of these four loci there is evidence for alternative signals at these loci in Europeans (we couldn't include the other ancestries here as we don't have relevant well-powered full summary statistics for these loci). This indicates that there may be alternative causal variants between ancestral groups at these loci or that the different LD structures can mask the same underlying causal variants. Exploring such discrepancies further in additional datasets will be an important direction for future work. We also include LD matrices of these loci (Supplementary Figure 3), which also support that for some loci, alternative LD patterns may contribute to the differences seen between ancestry groups for specific variants.

The additional text included related to this point is referred to in the response to comment 1.

- 2-2: The authors used MR-MEGA which accounts for the heterogeneity in the effect size estimates across ancestries. However, the substantially heterogeneous associations at rs77869365 and these four SNPs indicates MR-MEGA approach maybe too permissive about the heterogeneous signals. The authors should report statistics using an inverse variance-weighted fixed-effect meta-analysis and provide sufficient discussion on the discrepancies (if any) between MR-MEGA and fixed-effect meta-analysis approach.

Some of the concerns around these loci we believe are alleviated with the addition of the locus zoom plots as previously mentioned, but in our opinion the issue with these variants isn't the meta-analysis method used, but the fact that only one Japanese study is included and we lack replication

in this population. This is now explicitly stated in the discussion on line 486. Further response on the appropriateness of MR-MEGA are given in the response to comment 3.

- 3, Lines 282-283: “A further 4 SNPs did not replicate, and on closer examination (Supplementary Figure 2, and MAF in cases <1%), their association in the discovery analysis appeared to be driven by a false positive outlying result in a single European cohort.” This is also very concerning. This indicates that MR-MEGA approach may also be too permissive about the heterogeneous signals even within a single ancestry. How do the fixed-effect statistics behave at these SNPs? I expect that fixed-effect meta-analysis approach provides more conservative statistics (meaning less false positive signals) compared with MR-MEGA.

We agree that MR-MEGA (and indeed any other method that attempts to detect associations in a multi-ancestry setting) is sensitive to certain assumptions.

We also agree that the 4 results that proved to be false positives were not detected by the relevant fixed-effects analysis (European-only) - which was one of the results that helped confirm to us that these were not real. However, in that case we would argue that the approach of including replication was sufficient to determine that these were artefacts and so overall our approach is robust to this situation.

We have not undertaken a mixed ancestry fixed-effect analysis as conceptually this is the wrong approach, given the widely appreciated expectation for heterogeneity of effect in this situation. Random-effects analysis could have been used in this context, but that approach would be even more permissive of these heterogenous effects, with the strength of the MR-MEGA approach being that it only allows heterogeneity that is correlated with genetic ancestry. It has been shown in testing that MR-MEGA has appropriate false positive rates. However, the key limitation in **our use** of this method is that it is sensitive to being driven by single cohorts with unique patterns of genetic ancestry. However, our inclusion of a replication stage (for most variants) overcomes this. The only exception being the 4 potential Japanese-specific variants (for which we have no replication cohort). We therefore now state this particular limitation in the discussion:

“We report 4 previously identified loci (lead SNPs rs9864845, rs34665982, rs45602133, rs4312054) with associations that appear to be specific to the Japanese cohort (although driven by just one cohort and still require independent replication).”

- 4, The authors used ‘total evidence score’ to prioritize the candidate gene. I understand their motivation very well. However, any scoring system needs calibration. Their strategy is too arbitrary to be used in a scientific journal. They need to provide analyses such as those reported in the OpenTarget manuscript (PMID: 34711957). If they can’t provide sufficient analyses on this topic, they need to rely on previously-reported methods whose validity was confirmed in peer-reviewed journals.

The approach we have used in this study to prioritise candidate genes has been previously reported in a peer-review journal (Sobczyk et al, 2021, PMID 33901562).

We have now mentioned this in the methods section:

“To prioritise candidate genes at each of the loci identified in the European GWAS, we investigated all genes within +/- 500kb of each index SNP (selected to capture an estimated 98% of causal genes)⁶⁵. The approach used has been previously described by Sobczyk et al¹⁸.”

It would be great if we were able to undertake calibration (similar to that done by OpenTargets) as suggested, however, our pipeline is specific to eczema and in the OT gold standard list, only one gene is listed for eczema (at a locus that does not feature in our GWAS results). We acknowledge that our method is subjective, but our approach does identify genes previously implicated (and sometimes experimentally validated, e.g. *FLG* which has extensive in vitro validation). The method used also incorporates previously published methods and pipelines such as postGAP and OpenTargets, which at times gave conflicting results. We have reported output from these well-used pipelines (alongside our eczema specific analyses) in supplementary table 11 which the reader can interrogate further for loci of interest.

Minor comments:

- 1, The authors should report the number of cases in addition to the total sample size in the main text where they explain GWAS study design (e.g., L260 on page 8). Many recent GWASs have the control samples disproportionately more than case samples. If we only report total sample size, we might overemphasize the study scale.

We agree this is important to include. The numbers of cases was included in the methods, but we now also include them in the results section for clarity.

“The discovery European meta-analysis (N=864,982; 60,653 AD cases and 804,329 controls from 40 cohorts, summarized in Supplementary Table 2) identified 81 genome-wide significant independent signals (Figure 1a and Supplementary Figure 1).”

“In a multi-ancestry analysis including individuals of European, Japanese, Latino and African ancestry (Supplementary Table 2, N=1,086,394; 65,107 AD cases and 1,021,287 controls), a total of 89 signals were identified as associated with AD (Figure 1b and Supplementary Figure 1).”

- 2, Criteria of novel loci: The authors wrote “Novel loci are defined as a SNP that had not been reported in a previous GWAS (Supplementary Table1), or was not correlated ($r^2 < 0.1$) with a known SNP from this list. Which population did they use to calculate r^2 ? Do the authors calculate r^2 in the ancestry in which previous GWAS conducted?”

Yes, this is correct, we have altered the text for clarity:

“Novel loci are defined as a SNP that had not been reported in a previous GWAS (Supplementary Table 1), or was not correlated ($r^2 < 0.1$ in the relevant ancestry) with a known SNP from this list.”

3, Significant threshold:

- 3-1. At L264 (page 8), they set significant threshold at $6e-04$, which seems to be 0.05 divided by 81 (the number of target loci in this analysis). This makes sense but need to be explicitly explain in the main text (e.g., “ $P < 0.05/81$ tested loci ($=6e-04$)”). Similar modification should be applied to other related parts (e.g., L275 at page 9).

We have now made clear that the P -value thresholds that are based on Bonferroni corrected P -values

“All 81 were associated in the European 23andMe replication analysis (Bonferroni corrected $P < 0.05/81 = 6 \times 10^{-4}$), $N = 2,904,664$, Table 1).”

“70 of these had been detected in the European-only analysis and a further 12 showed some evidence for association (Bonferroni corrected $P < 0.05/89 = 5 \times 10^{-4}$) in the European analysis”

“Of the 19 loci that reached genome-wide significance in the multi-ancestry discovery analysis only (Table 3), 11 replicated in at least one of the replication samples (of European, Latino and/or African ancestry; Bonferroni corrected $P < 0.05/19 = 2 \times 10^{-3}$).”

- 3-2. How did the authors set the significant threshold in the conditional analyses to $1e-05$?

This is the default (and widely used) threshold for this analysis in GCTA-COJO software. This is now mentioned in the methods:

“COJO-slc was used to determine which SNPs in the region were conditionally independent (using default $P < 1 \times 10^{-5}$)”

- 4, Lambda is not an excellent metric to indicate the bias in GWAS results since this can be inflated by polygenic effects (biological effect), not necessarily by the bias. The authors should report the intercept of LDSC.

The intercept of the LDSC analysis has now been reported in the main text:

“The SNP-based heritability (h^2_{SNP}) for AD was estimated to be 5.6% in the European discovery meta-analysis (LDSC intercept=1.042 (SE=0.011)).”

- 5, In the Sup-Table 3, the authors should report the r^2 (in European reference panel) between the “Primary SNP” and the “Secondary SNP” to explain the variants are independent.

The r^2 between the primary and secondary SNP has now been reported in supplementary table 3.

- 6, Typo in Table3. At the row of rs7773987 the authors put “***” and the legend says it is about “rs77869365”, a different SNP.

Thank you for spotting this mistake. The correct SNP is rs77869365 which has an opposite direction of effect in Europeans compared to Japanese individuals. The table has been edited so the ** are

now included on the correct row.

REVIEWER COMMENTS

Reviewer #1 (Remarks to the Author):

The revised manuscript has addressed many of the concerns raised by the reviewers, and the overall approach and findings are sound. The results, however, still seem better suited for another journal focused on the specifics of association results in AI or to wait until additional confirmatory translational findings are completed and added.

Reviewer #2 (Remarks to the Author):

Summary:

Aggrey et al generally addressed my concerns, but several concerns have not been fully addressed.

Major comments:

1, Opposite allelic directions between Europeans and Japanese at rs77869365.

They stated "we now include a more thorough investigation of all ancestry-inconsistent results (including this SNP), showing the locus zoom plot and LD matrices for these loci" in the response letter. However, just providing these plots do not at all explain WHY the effect size directions are discordant. Their response is just literally providing the plot in a very naïve way and adding general speculations such as LD structure differences are the culprit. Their explanations are so rough and failed to convince the readers. They need to provide additional analyses that explain the potential reasons for this discrepancy. For example, is LD structure ancestral difference more pronounced at the loci with inconsistent effect sizes than at loci with consistent effect sizes?

2, Japanese specific SNPs (rs9864845, rs34665982, rs45602133, rs4312054)

2-1: Similar to comment 1, for these SNPs, they only provided locus zoom plots and LD matrix. They did not provide any "analysis" to interpret the ancestry gap in the association.

2-2: The fact that rs34665982 is within the HLA region is super concerning. This region has a strong and long-range LD structure, requiring special consideration. Also, HLA coding variants probably play a role at this locus. If they can't provide sufficient analyses for the HLA region, I strongly recommend simply excluding the HLA region. Otherwise, it is very misleading for readers.

3. The use of MR-MEGA

Their choice of MR-MEGA is understandable. However, as they admit in the response letter MR-MEGA "only allows heterogeneity that is correlated with genetic ancestry". I agree that "MR-MEGA has appropriate false positive rates" but this is the case only in an ideal situation. Do they have evidence that MR-MEGA p values are well-calibrated specifically in their study?

4. No single method is perfect. Most of the time, each method has its pros and cons. MR-MEGA approach has its own limitation as discussed in the response letter. Fixed effect meta-analysis is also not perfect, but it has fewer false positives (just loses power), also we can estimate heterogeneity across ancestry. I recommend reporting both statistics of MR-MEGA and fixed-effect meta-analyses. In the original MR-MEGA article, they reported both statistics.

Response to Reviewers

Reviewer #1 (Remarks to the Author):

The revised manuscript has addressed many of the concerns raised by the reviewers, and the overall approach and findings are sound. The results, however, still seem better suited for another journal focused on the specifics of association results in AI or to wait until additional confirmatory translational findings are completed and added.

We would contend that several aspects of this GWAS and follow-up analysis represent state-of-the-art analytical procedures that will be of wide interest to readers with interest in genetic epidemiology of any trait. We assume that the editor is willing to consider our manuscript for publication in Nature Communications.

Reviewer #2 (Remarks to the Author):

Summary:

Aggrey et al generally addressed my concerns, but several concerns have not been fully addressed.

Following this reviewer's very helpful comments, we have made several changes and additions to the multi-ancestry analysis, which address their remaining concerns:

- We now have a more stringent procedure for identifying the multi-ancestry results which are independent of the European signals. Now, for multi-ancestry index variants within 500kb of index SNPs identified in the European-only analysis, we considered these to be independent (and hence reported them) if the lead multi-ancestry SNP was not in LD ($r^2 < 0.01$) with the lead neighbouring European variant. This means 5 loci drop out of Table 3, including 3 mentioned in the below comments.
- We now also add multi-ancestry fixed effect meta-analysis for comparison with the MR-MEGA results (results given in Supplementary Table 4). Whilst we are not able to access replication results for any alternative SNPs which reach the index SNP position with this method, we do report in the text where the fixed effect analysis highlights interesting alternative SNPs for the "Japanese-specific" loci.
- We have replaced our previous supplementary Figure 3, which showed LD matrix plots that were unhelpful, with a complete set of locus zoom plots (multi-ancestry MR-MEGA, multi-ancestry fixed effects, European and Japanese) for better visualisation of the effects across ancestries in the regions with inconsistent effects. To better explore ancestry-inconsistent results in general we also add a supplementary Figure 4 which plots the effects of all SNPs across all ancestries. We have coloured this plot to highlight SNPs showing a seeming difference (i.e. non-overlapping confidence intervals), but interestingly, most of those that show an apparent difference do not feature as outliers on these plots, suggesting the differences in effects reported are likely mostly due to chance.

Major comments:

1, Opposite allelic directions between Europeans and Japanese at rs77869365.

They stated “we now include a more thorough investigation of all ancestry-inconsistent results (including this SNP), showing the locus zoom plot and LD matrices for these loci” in the response letter. However, just providing these plots do not at all explain WHY the effect size directions are discordant. Their response is just literally providing the plot in a very naïve way and adding general speculations such as LD structure differences are the culprit. Their explanations are so rough and failed to convince the readers. They need to provide additional analyses that explain the potential reasons for this discrepancy. For example, is LD structure ancestral difference more pronounced at the loci with inconsistent effect sizes than at loci with consistent effect sizes?

Following the above changes, this SNP (showing an opposite effect between Europeans and Japanese) now drops due to our more stringent independence test (i.e. this SNP is not independent of rs112385344, which is ~19kb away, $r^2=0.26$ in Europeans, monomorphic in Japanese).

Interestingly when we ran the fixed effects MA, as suggested by the reviewer, an alternative top SNP is discovered in this region (rs62162286) and this SNP showed a consistent direction of effect. So we still suspect that LD differences explain the discrepancy we saw, but nevertheless we now drop this SNP for reasons given above.

2, Japanese specific SNPs (rs9864845, rs34665982, rs45602133, rs4312054)

2-1: Similar to comment 1, for these SNPs, they only provided locus zoom plots and LD matrix. They did not provide any “analysis” to interpret the ancestry gap in the association.

It is not apparent what additional analysis this reviewer would suggest, but for the two remaining purportedly Japanese-specific loci we now show through fixed effect meta-analysis that alternative index SNPs have large discrepancies in allele frequency between Europeans and Japanese (~34% and 13% in Japanese, but both <1% in Europeans).

One of these (rs59039403) was previously suggested to be the functional variant given it is a deleterious missense mutation. The other locus was previously further investigated with functional follow-up and a putative functional variant (rs12637953) identified given its role as an enhancer for CCDC80 promoter. Therefore, in the absence of replication we believe this additional information increase the likelihood that these are real Japanese-specific variants for AD, and we now include thorough results and discussion for these two loci in the manuscript.

2-2: The fact that rs34665982 is within the HLA region is super concerning. This region has a strong and long-range LD structure, requiring special consideration. Also, HLA coding variants probably play a role at this locus. If they can't provide sufficient analyses for the HLA region, I strongly recommend simply excluding the HLA region. Otherwise, it is very misleading for readers.

With our more stringent independence test the HLA signal is now excluded from the multi-ancestry loci.

3. The use of MR-MEGA

Their choice of MR-MEGA is understandable. However, as they admit in the response letter MR-MEGA “only allows heterogeneity that is correlated with genetic ancestry”. I agree that “MR-MEGA has appropriate false positive rates” but this is the case only in an ideal situation. Do they have evidence that MR-MEGA p values are well-calibrated specifically in their study?

We have followed the recommendation to undertake a fixed effect meta-analysis and now also report these results to allow full interpretation. MR-MEGA does show higher genomic inflation than the fixed effects model (MR-MEGA $\lambda=1.12$, fixed effects $\lambda=1.04$).

4. No single method is perfect. Most of the time, each method has its pros and cons. MR-MEGA approach has its own limitation as discussed in the response letter. Fixed effect meta-analysis is also not perfect, but it has fewer false positives (just loses power), also we can estimate heterogeneity across ancestry. I recommend reporting both statistics of MR-MEGA and fixed-effect meta-analyses. In the original MR-MEGA article, they reported both statistics.

We now report both MR-MEGA and fixed effect results, as suggested.

REVIEWERS' COMMENTS

Reviewer #2 (Remarks to the Author):

The authors fairly addressed all of my previous concerns. I do not have any further comments.